# Position: Probabilistic Modelling is Sufficient for Causal Inference

**Bruno Mlodozeniec** [1 2]  **David S. Krueger** [3]  **Richard E. Turner** [1 4]

## Abstract

Causal inference is a key research area in machine learning, yet confusion reigns over the tools needed to tackle it. There are prevalent claims in the machine learning literature that you *need* a bespoke causal framework or notation to answer causal questions. In this paper, we want to make it clear that you *can* answer any causal inference question within the realm of probabilistic modelling and inference, without causal-specific tools or notation. Through concrete examples, we demonstrate how causal questions can be tackled by *writing down the probability of everything*. Lastly, we reinterpret causal tools as emerging from standard probabilistic modelling and inference, elucidating their necessity and utility.

## 1. Introduction

Causal inference questions play a key role in many areas of machine learning (Kaddour et al., 2022), such as policy (Athey & Imbens, 2017), healthcare (Sanchez et al., 2022), or fairness (Kusner et al., 2017). Despite the growing emphasis on these problems within the machine learning research community, there is still an apparent lack of clarity regarding the tools and frameworks necessary to tackle them. Concretely, there is a seeming prevalent belief that one cannot answer causal questions without adopting tools from outside machine learning and probability.

This lack of clarity is epitomised by the disagreement between Judea Pearl and Andrew Gelman — foundational researchers in the area of causality and statistical inference who appear to hold contradictory views on the technical foundations of their field. Whereas Pearl holds that *"there is no way to answer causal questions without snapping out of statistical vocabulary"* (Hartnett, 2018) and that *"we need to enrich our language with a do-operator"* (Pearl,

---

[1]University of Cambridge [2]Max Planck Institute for Intelligent Systems, Tübingen [3]MILA [4]The Alan Turing Institute. Correspondence to: Bruno Mlodozeniec <bkm28@cam.ac.uk>.

*Proceedings of the 42ⁿᵈ International Conference on Machine Learning*, Vancouver, Canada. PMLR 267, 2025. Copyright 2025 by the author(s).

2019a), Andrew Gelman states: *"I find it baffling that Pearl and his colleagues keep taking statistical problems and, to my mind, complicating them by wrapping them in a causal structure"* (Gelman, 2019).

Such apparent allegations of insufficiency of standard statistical tools — such as probabilistic modelling or Bayesian Networks — are perpetuated in the machine learning literature. Claims that causal questions *"[cannot] be answered with statistical tools alone, but require methods from causality"* (Pawlowski et al., 2020) are prevalent (see Appendix D). Although often intended as a forewarning of the dangers of *naïvely* applying probabilistic tools to solve causal problems — which we demonstrate in Section 2.1 — these comments leave many confused and sceptical of the role of probabilistic modelling in solving causal problems.

In this paper **we show that the standard tools of probabilistic modelling *are* sufficient for causal inference**. In particular, we show that one just needs to follow the 'one' overarching rule advocated by David MacKay:

> *"Always write down the probability of everything."*
> — Steve Gull (MacKay, 2003, p. 61)

We argue that **the resulting approach is *clear*, *unifying*, and *general for answering causal inference questions***.

Heeding Pearl (2019b)'s call for concrete examples, we will demonstrate the probabilistic approach on simple examples of causal inference problems – in turn, interventional (Section 2) and counterfactual (Section 3). For each, we will illustrate how one can solve it by 'writing down the probability of everything'. There are convenient classes of models (e.g. Structural Causal Models (Peters et al., 2017)), useful notational shorthands (e.g. the *do*-operator), and a machinery developed around them (e.g. the *do*-calculus (Pearl, 2009)) to tackle causal questions, which we will introduce as "syntactic sugar" in the probabilistic framework.

We contend that the entire debate hinges on a simple semantic confusion: the term "statistical" is often used too narrowly to refer only to methods for modelling correlations or associations among the observed variables. This restricted definition is the basis for claims that statistics or probability are insufficient for causality (Appendix D). In Appendix Q, however, we argue this terminological choice is not only confusing, but also historically and substantively inaccurate.

While the warnings against a *naïve* probabilistic approach are well-founded, the conflation of the restricted 'associational' approach with the entire framework ultimately muddies the waters and fuels the very debate we seek to resolve.

We advocate for the position that **the probabilistic modelling approach is useful, yet, under-recognised**. It makes causal inference accessible to a large part of the machine learning community and lowers the barrier to entry to tackling causal problems by removing the need for familiarity with a specialised notational toolkit. Secondly, probabilistic modelling is often a more flexible and general tool, making it a promising vehicle for future causality research Section 4.1.

As this position can easily be misconstrued, we want to clarify: We claim the causal toolbox is *not necessary* for causal inference, but not that it's without utility. We are not seeking to understate the many insights regarding the challenging nature of causal inference problems that have come from Pearl and the causal inference community. Neither are we arguing that the mainstream (Pearl's) causal framework and notation should be disbanded with. We are also not proposing a new framework. Our goal is to clearly illustrate how causal problems can be tackled in an existing one.

## 2. Interventions

Throughout this paper, we will use the running example of aspirin's efficacy on headache duration to ground the discussion in a concrete problem. In this section, we will first introduce the problem, the data generating process, and a causal (interventional) question. We will illustrate how a probabilistic modelling approach where we write down the joint of everything can be applied, and finally relate it to the "classical" causal approach.

### 2.1. A Concrete Example

Let's start by specifying a model for the data-generating process in the observed setting.[1]

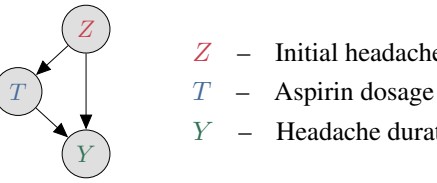

| | | |
|---|---|---|
| $Z$ | – | Initial headache severity |
| $T$ | – | Aspirin dosage |
| $Y$ | – | Headache duration |

*Figure 1.* Graphical model for the observational data on aspirin's effect in a population.

In the model, we will assume that every variable is

---

[1]We denote random variables with capital letters (e.g., $X$), the values in the space of possible realisations with lowercase letters (e.g., $x$). We write $p_{X,Y}(x,y)$ to denote a probability density/mass function of $X, Y$, or $p(x,y)$ whenever the random variables are apparent from context. Sometimes we abbreviate $p(\cdot \mid \boldsymbol{\theta})$ to $p_{\boldsymbol{\theta}}(\cdot)$ to signify that the distribution is parameterised by $\boldsymbol{\theta}$.

distributed according to a log-normal distribution (Appendix A.1) like so:

$$Z \sim \log\mathcal{N}\left(\mu_Z, \sigma_Z^2\right)$$
$$T = Z^a \varepsilon_T \qquad \text{where } \varepsilon_T \sim \log\mathcal{N}\left(0, \sigma_T^2\right) \quad (1)$$
$$Y = \frac{Z^b}{T^c}\varepsilon_Y \qquad \text{where } \varepsilon_Y \sim \log\mathcal{N}\left(0, \sigma_Y^2\right)$$

Here, $\theta = \{a, b, c, \mu_z, \sigma_Z, \sigma_T, \sigma_Y\}$ are the parameters to be inferred from the observed data.

As seen in Equation (1), the parameter $c$ captures the strength of the effect of aspirin dosage on the headache duration; the larger the value of $c$, the more effective aspirin is. Similarly, $a$ captures how the headache severity affects the decision to take a given dose of aspirin, and $b$ controls how the headache severity affects the headache duration. We assume $a, b > 0$ to reflect that higher initial headache severity makes people take *more* aspirin, and makes the headache last *longer*. From the properties of log-normal distributions, it follows that $\begin{bmatrix} Z & T & Y \end{bmatrix}^\mathsf{T} \sim \log\mathcal{N}\left(\boldsymbol{\mu}, \boldsymbol{\Sigma}\right)$ are jointly distributed according to a multivariate log-normal (see Appendix B.1).

---

> **Example 1:** **Interventions on Aspirin Dose**
>
> **Inference question** In the observed world, in which the subjects decide their own aspirin dosage, we've surveyed people on their headaches to collect a dataset $\mathcal{D} = \{(z_i, t_i, y_i)\}_{i=1}^N$. We want to use this dataset to answer the question: what is the expected effect of intervening to assign someone a given dose of aspirin $t^*$ on their headache duration on average?

---

This question can be considered an instance of a "causal" inference question, as it asks what would happen in a hypothetical setting where we have altered the mechanism by which the aspirin dose is determined.

One has to take care when approaching such problems — as has clearly been demonstrated by e.g. Pearl (2009) — due to *confounding*. In particular, a naïve supervised learning approach of estimating the conditional distribution $p_{Y|T}(\cdot \mid t^*)$ to compute the conditional expectation $\mathbb{E}[Y|T = t^*]$ would not give the right answer to the question that we asked. The conditional expectation $\mathbb{E}[Y|T]$ can be computed analytically for this problem (Appendix B.2). In the special case when $c = 0$ — i.e. aspirin has no effect on the headache duration ($Y = Z^b \varepsilon_T$) — this reduces to:

$$\mathbb{E}[Y|T] = \exp\left(\boxed{\frac{ab\sigma_Z^2}{\sigma_Z^2 + \sigma_Y^2}\left(T - a\mu_z\right)} + \text{const.}\right)$$

positive whenever $ab > 0$

In this case, we observe that the expected headache duration goes up with the ingested aspirin dose $T$, even though

aspirin dose has no effect on the headache duration ($c=0$). Even when $c > 0$ — i.e. when aspirin has a remedial effect on the headache duration — it's possible to observe that people who take a higher aspirin dose have a longer headache on average (see Equation (9)). The longer headache in those who had taken a higher dose is, of course, due to a more severe initial headache, rather than due to adverse workings of aspirin. Figure 2 illustrates this phenomenon, which is often referred to as the Simpson's paradox (Simpson, 1951; Pearl, 2009, §6.1).

The inadequacy of the naïve approach to answering such inference questions is the cornerstone of arguments that machine learning techniques are not sufficient for causal inference without some enrichment by a causal framework (Pearl, 2009; Pearl & Mackenzie, 2018). It certainly highlights that inference corresponding to causal questions often requires generalising beyond the observed data distribution, and that — as we will illustrate below — significant modelling assumptions are necessary to answer causal questions. However, we will show precisely how these assumptions can be specified by defining a joint model over all the tasks or settings that we care about.

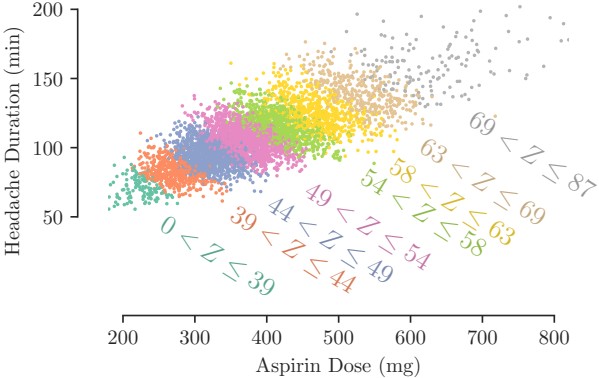

*Figure 2.* Samples of the headache duration against aspirin dose from the log-normal aspirin model. The plot shows headache duration increasing with aspirin dose taken; however, for any group of people with a narrow range of headache severities, the trend is reversed. Parameters for the model are $a=1.5$, $b=2.68$, $c=1.0$, $\mu_Z=3.95$, $\sigma_Z=0.15$, $\sigma_T=0.07$ and $\sigma_Y=0.05$.

### 2.2. Modelling Interventions with Bayesian Networks

To isolate the effect of aspirin, we are asking what would happen if we hypothetically *intervened* to assign people a higher dose. Such an intervention would clearly change the mechanism by which a person's dose is determined; the probability distribution over the dose conditioned on the headache severity in the hypothetical intervened-upon setting would be different.

Let's consider how we could model such an intervention.

In a probabilistic framework, *the first step in any inference procedure is to write down all the assumptions about the problem*. From these assumptions, a joint distribution over all variables and settings of interest should follow. That joint distribution can then be queried for any quantities of interest. Below, we're going to follow through with this approach, and write a joint distribution over the original *unintervened* world and the new *intervened* world by first specifying exactly how they are related.

To this end, in addition to the random variables corresponding to the observed dataset $\{(Z_i, T_i, Y_i)\}_{i=1}^{N}$, which we assume has been generated as described in eq. 1, we also define variables $Z^*, T^*, Y^*$, which correspond to the intervened-upon setting. For notational convenience, we'll write $q(\cdot)$ in place of $p(\cdot)$ when referring to distribution functions over interventional variables $Z^*, T^*, Y^*$ when using the machine learning short-hand; for instance, we'll write $q(y|t, z)$ in place of $p_{Y^*|T^*, Z^*}(y|t, z)$.

---

***Example 1 continued:***

**Model and assumptions**  Let's begin by setting up a model in which the two settings of interest — the observed and the interventional — are modelled explicitly. We're going to make the assumptions embodied in the Bayesian Network (Appendix A.2) in the figure below:

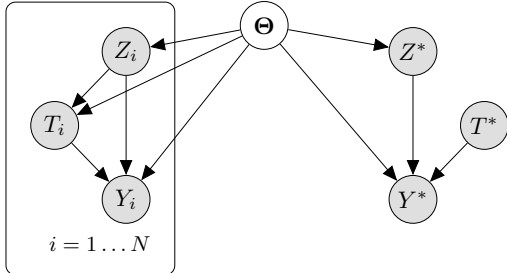

*Figure 3.* Graphical model for interventional aspirin example.

Parameters $\Theta$ are shared between the two worlds allowing observations in the real world to inform inference in the intervened-upon one. The aspirin dose $T^*$ in the interventional setting is independent of the headache severity $Z^*$, reflecting that we are asking a question about the effect of a hypothetical assignment of a given dose regardless of the subject's initial headache severity.

The graph above specifies independence assumptions, but we need to further specify the distribution $q(\cdot)$ on the variables in the intervened-upon setting. We assume that the physical mechanism determining the headache duration $Y^*$ based on the initial headache severity $Z^*$ and aspirin dose ingested $T^*$ remains unchanged in the interventional setting; in other words, the physiological

properties of the subjects in the hypothetical setting are the same as in the observed one. Hence, we assume that the conditional distribution of the headache duration $Y^*$ conditioned on $T^*, Z^*$ in the intervened-upon world is the same as the distribution of $Y$ conditioned on $T, Z$ in the observed world: $p_{\boldsymbol{\theta}}(y|t, z) = q_{\boldsymbol{\theta}}(y|t, z)$.

To represent the belief that we are manipulating the system to assign a specific dose $t^*$ independently of $Z^*$, we specify $q_{\boldsymbol{\theta}}(t) = q(t) = \delta(t^* - t)$.

Lastly, the distribution on the initial headache severity $Z^*$ in the intervened-upon setting is assumed to be the same as in the observed setting: $q_{\boldsymbol{\theta}}(z) = p_{\boldsymbol{\theta}}(z)$.

Based on the above assumptions and graphical model, we can write down a full joint distribution:

$$p(z^*, t^*, y^*, \boldsymbol{\theta}, \mathcal{D}) = \overbrace{p_{\boldsymbol{\theta}}(z^*)q(t^*)p_{\boldsymbol{\theta}}(y^*|z^*, t^*)}^{\text{Interventional world}}$$
$$\times \underbrace{\left( \prod_{i=1}^{N} p_{\boldsymbol{\theta}}(z_i)p_{\boldsymbol{\theta}}(t_i|z_i)p_{\boldsymbol{\theta}}(y_i|z_i, t_i) \right)}_{\text{Observed world}} \times p(\boldsymbol{\theta}) \quad (2)$$

See Appendix B.3 for a step-by-step derivation where we highlight which of the above assumptions were used at each step.

Given a joint, probability theory tells us how to find all the marginal distributions, conditional distributions and expectations of interest.[2]

---

**Example 1 continued:**

**Inference** Returning to the inference question, we wanted to know the expected headache duration in the intervened world in response to a given dose $t^*$ (which is a fixed parameter of the model). Hence, we compute the expected value of $Y^*$ given the observed data $\mathcal{D}$ (see Appendix B.4):

$$\mathbb{E}[Y^*|\mathcal{D}] = \mathbb{E}\left[ \mathbb{E}[Y^*|\Theta] \mid \mathcal{D} \right] \quad (3)$$

$$= \iiint \underbrace{y^* q_{\boldsymbol{\theta}}(y^*|z^*, t^*)q_{\boldsymbol{\theta}}(z^*)dz^*dy^*}_{\substack{\text{Calculate expectation in the intervened} \\ \text{world conditioned on posterior over } \theta}} \underbrace{p(\boldsymbol{\theta}|\mathcal{D})d\boldsymbol{\theta}}_{\substack{\text{Infer model parameters} \\ \text{using observed data } \mathcal{D}}}$$

Note that in Equation (3), all the density functions in the inner integral are the same as these in the observed distribution. For the log-normal aspirin model, we can actually calculate the inner expectation exactly (Appendix B.5):

$$\mathbb{E}[Y^*|\boldsymbol{\theta}] = (t^*)^{-c} \exp\left( b\mu_Z + \frac{b^2\sigma_Z^2 + \sigma_Y^2}{2} \right) \quad (4)$$

---

As expected, the effect of assigning a different dose of aspirin $t^*$ is only determined by the parameter $c$; whenever $c > 0$, intervening to administer a higher dose will in expectation shorten the headache duration.

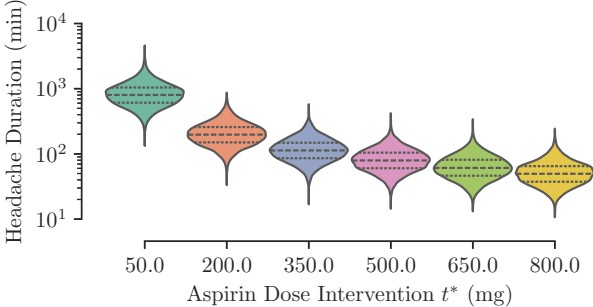

*Figure 4.* Distributions of the headache duration in the intervened upon world for different interventions $t^*$ on the assigned aspirin dose in the log-normal aspirin model.

In the example above, we demonstrated a probabilistic modelling approach to solving a causal inference problem. We showed how interventional questions can be answered by specifying a joint model over both the intervened-upon and observed settings. We used a Bayesian Network (BN) to encapsulate assumptions about both worlds and how they are related. We refer to this approach as the *twin model* approach.[3] This method works; the assumptions conveyed are transparent, and as long as they are correct, it's a valid way to approach causal inference. Crucially, we didn't need any bespoke causal syntax (e.g. the *do*-notation) to do so[4]. We summarise this approach in Appendix E. Of course, when specifying the assumptions, the devil is in the details. In Appendix F, we highlight the challenges that arise when constructing a model for interventional problems that practitioners need to be weary of.

### 2.3. "Classical" Approach: Causal Bayesian Networks

We will now relate the twin model approach to the more conventional presentation of causal tools. Specifically, we will introduce the intervention operation and the *do*-notation as a concise syntax for specifying a joint model over the observed and interventional settings. With this approach, we will specify a graph over a single set of variables in the observed world only — like the one over $Z, T, Y$ in Figure 1 — in a way that encompasses the assumptions about how to extend it to other interventional settings of interest. Namely, we will specify a *rule* for obtaining a joint distribution in the intervened setting from a graphical model of the observed setting.

---

[2]How to actually estimate the numerical values for these expressions then falls right within the realms of machine learning and statistics. See Appendix G for a discussion.

[3]As it is closely related to the *twin network* method presented by Balke & Pearl (1994).

[4]Note that the graphical model in Figure 3 is a Bayesian network (as defined in Def. A.3) – a strictly probabilistic tool.

Formally, we can define an *intervention operation* on a BN as follows:[5]

**Definition 2.1. Interventions in Bayesian Networks.** Consider a Bayesian Network on $\mathbf{X} = \{X_1, \ldots, X_d\}$ with a graph $\mathcal{G}$ that entails the factorisation $p(\mathbf{x}) = \prod_{i=1}^{d} p(x_i|\mathbf{x}_{\mathbf{PA}_i^{\mathcal{G}}})$. An *intervention* on node $j$ with new parents $\mathbf{PA}_j^*$ and a new conditional distribution $g^*(x_j|\mathbf{x}_{\mathbf{PA}_j^*})$ yields a new BN on $\mathbf{X}^* = \{X_1^*, \ldots, X_d^*\}$ with a graph $\mathcal{G}^*$. The graph $\mathcal{G}^*$ is identical to $\mathcal{G}$ with the exception that the edges going into $j$ are replaced with a new set of edges from the new set of parents $\mathbf{PA}_j^*$. The intervened-upon BN has a new entailed joint distribution:

$$q(\mathbf{x}) = g^*(x_j|\mathbf{x}_{\mathbf{PA}_j^*}) \prod_{i \in \{1,\ldots,d\}\setminus\{j\}} p(x_i|\mathbf{x}_{\mathbf{PA}_i^{\mathcal{G}}}) \quad (5)$$

Here, the new conditional distribution function $q(x_j|\mathbf{x}_{\mathbf{PA}_j^*}) = g^*(x_j|\mathbf{x}_{\mathbf{PA}_j^*})$ replaces the previous one $p(x_j|\mathbf{x}_{\mathbf{PA}_j^{\mathcal{G}}})$ in the factorisation, while the remaining conditionals are kept the same.

OBSERVED WORLD BN      INTERVENED-UPON BN

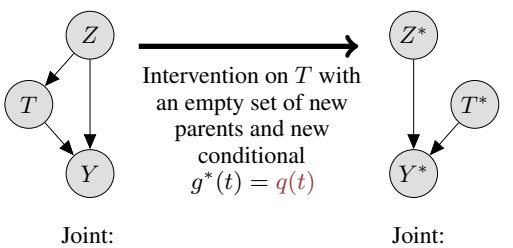

Joint:
$p(z,t,y) = p(z)p(t|z)p(y|t,z)$    $q(z,t,y) = p(z)q(t)p(y|t,z)$

*Figure 5.* Illustration of the rule in def. 2.1 for obtaining the joint on the intervened-upon variables from the BN over the observed variables.

There is an intuitive connection between the graph in a BN, and the everyday meaning of the terms 'cause' and 'effect', as has been noted by Pearl (2009). If we specify the graph such that the distributions obtained through applying the intervention operator match up with what we would *expect* to happen were we to perform interventions in the real system, the edges in the graph would match up with what we would colloquially call causal relationships.

***do*-notation for interventions** The *do*-notation can be used as a shorthand for denoting the result of applying the intervention operation. For instance, one could write $q(\mathbf{x}) = p_{do(X_k := g^*(\cdot|x_{\mathbf{PA}_k^*}))}(\mathbf{x})$ to denote an intervention on node $k$ with a new set of parents $\mathbf{PA}_k^*$ and a new conditional distribution given parents $g^*(\cdot|x_{\mathbf{PA}_k^*})$. For atomic

---

interventions, the notation is often simplified further to $p(\mathbf{x}|do(X_k = c)) = p_{do(X_k := \delta(\cdot - c))}(\mathbf{x})$. The conditioning-like syntax in this notation can be seen as signifying that this joint is equal to the distribution in a Randomised Controlled Trial conditioned on $X_k = c$ (Appendix I).

The above machinery, built on top of probability theory, leads to some conveniences. For example, it leads to a succinct definition of *confounding*: the effect of $T$ on $Y$ is said to be confounded whenever $p(y|t) \neq p(y|do(T=t))$.

***do*-calculus** Once the model is defined, we can use standard probability theory to obtain any quantities of interest from the joint. However, in the context of causal inference, there are some common operations and, consequently, shortcuts that might be deployed. Various "adjustment criteria" and the *do*-calculus have been developed to address issues in that domain, arguably providing utility to a practitioner familiar with them. In Appendix J, we discuss these tools in more detail, and illustrate how they build on top of the language of probabilistic modelling and inference.

### 2.3.1. EQUIVALENCE CLASSES & DEFINITIONS

The intervention is a purely mathematical operation on a BN and its graph; it gives us a shorthand syntax for defining distributions over multiple settings of interest. We could choose to parameterise the conditionals $p(x_i|\mathbf{x}_{\mathbf{PA}_i^{\mathcal{G}}})$ with parameters $\boldsymbol{\theta}$. If we were to treat the parameters $\boldsymbol{\Theta}$ probabilistically, we could further assume that all the variables in settings derived by applying the intervention operator in Def. 2.1 are conditionally independent given $\boldsymbol{\Theta}$. This would allow us to define the exact model in Example 1 using this shorthand notation.

However, one has to be careful when defining a joint model with a BN and the intervention operator, because of *Markov Equivalence Classes*. In the traditional way of deploying Bayesian Networks (such as in the *twin model* approach), one does not have to worry about this; the role of the graph is strictly to specify a set of conditional independencies. If multiple graphs entail the same set of conditional independencies — i.e they are said to be part of the same *Markov Equivalence Class* — a practitioner can pick between them at will to enforce the same independence constraints. In this new way of deploying Bayesian Networks — by using the shorthand graph and the intervention operation in Def. 2.1 on the graph — different graphs in the same Markov Equivalence Class can entail different joint distributions. In this context, which graph from the equivalence class is chosen *does* make a difference (see Appendix H for details), and so one needs to carefully pick between them.

---

[5]Here, we continue to use the notation $q(\cdot)/p(\cdot)$ when referring to density or mass functions in the intervened/observed setting.

## 3. Counterfactuals

In the previous section, we showed one can tackle interventional questions by writing down the probability of everything. In this section, we are going to follow the same approach to answer counterfactual questions. We will end up with a structure that is similar to that in Example 1, but one that shares *individual* rather than *group* variables across the two settings of interest. We will start again by giving a concrete example of a counterfactual inference question, illustrate how to answer it within the probabilistic modelling framework, and then relate the approach to other causal graphical frameworks.

---

*Example 2:* **Aspirin Model Counterfactual**

**Inference problem** A friend tells you that she had a headache before her exam. She says that she took a given dose of aspirin $t$, and the headache lasted for $y$ hours, disrupting her exam performance. She doesn't recall the initial headache severity. You have access to the survey dataset $\mathcal{D} = \{(z_i, t_i, y_i)\}_{i=1}^{N}$ from Example 1. You are now wondering: had your friend taken some larger dose of aspirin, would her headache had gone away before the exam?

**Model and assumptions** Let's again set up a model over all the settings of interest — the observed and the counterfactual. In addition to the random variables $Z, T, Y$ in the observed setting, where $Z$ is now latent, we also define variables $T^*$ and $Y^*$, which correspond to the counterfactual dose choice and headache duration respectively. We assume that $Z$ — the headache severity — is the same in the counterfactual and the observed world. In addition, we still have the variables for the dataset of previous fully-observed cases $\{(Z_i, T_i, Y_i)_{i=1}^{N}\}$, which we can use to infer the parameters of our model. We'll use $q(\cdot)$ again to refer to densities in the counterfactual world, e.g. $p_{Y^*|T^*,Z}(y|t,z) = q(y|t,z)$ or $p_{T^*}(t) = q(t)$.

Again, we can further encode assumptions about how the variables are related with a Bayesian Network:

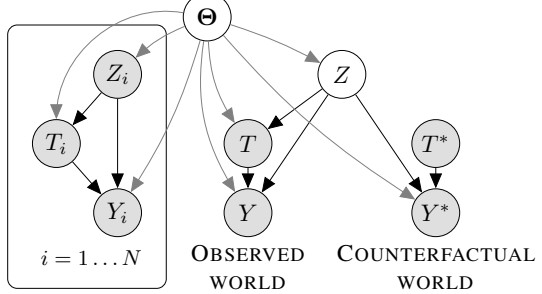

*Figure 6.* Graphical model for counterfactual inference in the aspirin example.

---

In the hypothetical counterfactual world, our friend no longer decides what dose to take based on the initial headache severity $Z$ — we've intervened to decide for her. To reflect this assumption, we remove the edge from $Z$ to $T^*$ in the counterfactual setting, making $Z$ and $T^*$ marginally independent,[6] and set $q(t) = \delta(t^* - t)$. Again, we assume $q_{\boldsymbol{\theta}}(y|t, z) = p_{\boldsymbol{\theta}}(y|t, z)$.

Based on the graphical model, we can again write down a full joint distribution over the settings of interest (Appendix B.6):

$$
\underset{\text{Dataset (joint) density}}{
p(z, t, y, t^*, y^*, \boldsymbol{\theta}, \mathcal{D}) = \prod_{i=1}^{N} p_{\boldsymbol{\theta}}(z_i) p_{\boldsymbol{\theta}}(t_i|z_i) p_{\boldsymbol{\theta}}(y_i|z_i, t_i)}
$$

$$
\times \Big( \underset{\text{Observed world}}{p_{\boldsymbol{\theta}}(t|z) p_{\boldsymbol{\theta}}(y|t, z)} \quad \underset{\text{Shared in both}}{p_{\boldsymbol{\theta}}(z)} \quad \underset{\text{Counterfactual world}}{q(t^*) p_{\boldsymbol{\theta}}(y^*|t^*, z)} \Big) p(\boldsymbol{\theta})
$$

**Inference** We wanted to infer the likely headache duration in the counterfactual world had we given our friend $t^*$ milligrams of aspirin. In the model, we can phrase this question as the conditional probability distribution $p(y^*|t, y, \mathcal{D})$ — the distribution over the hypothetical headache duration in response to aspirin dose $t^*$, given that we've observed that the actual headache lasted for $y$ minutes after our friend took a dose $t$, and given the dataset $\mathcal{D}$:

$$
p(y^*|t, y, \mathcal{D}) = \mathbb{E}_{p(\boldsymbol{\theta}|\mathcal{D}, t, y)} \left[ \int p_{\boldsymbol{\theta}}(y^*|t^*, z) p_{\boldsymbol{\theta}}(z|t, y) dz \right]
\tag{6}
$$

Again, we can obtain analytical formulas for the inner integral in Equation (6) or for the conditional expectation $\mathbb{E}[Y^*|T, Y, \boldsymbol{\theta}]$. Coupled with a MAP or a maximum-likelihood estimate of the parameters $\boldsymbol{\theta}$, we can answer our counter-factual query (see Appendix B.7 for details).

---

The inner expression in Equation (6) can be interpreted as computing the marginal over $Y^*$ in an intervened-upon model where distribution for the headache severity has been replaced by the posterior on $Z$ conditioned on the observed data. This observation is useful for connecting this approach to Structural Causal Models. See Appendix L for more details. We'll discuss this observation shortly.

### 3.1. Which Latent Variables Should We Share?

In Example 2, we considered a counterfactual world in which our friend had the same headache severity as in the observed world. Headache severity is the only variable shared

---

[6]Note that, if we didn't remove the $Z \to T^*$ edge the choice of a hypothetical dose $T^*$ in the counterfactual world would influence our belief over the headache severity $Z$ in the observed world, which would be somewhat paradoxical.

between the observed and the hypothetical world. However, you may recall that in our original formulation of the model in eq. 1 we specified $Y = \frac{Z^b}{T^c}\varepsilon_Y$. We used this equation as a shorthand to define that $Y$ conditioned on $T, Z$ is log-normal distributed: $Y|Z, T \sim \log\mathcal{N}(\frac{Z^b}{T^c}, \sigma_Y^2)$. So far in this note, we've only made use of this conditional distribution, and in no way relied or considered the dependence on $\varepsilon_Y$. Should we have shared $\varepsilon_Y$ between the counterfactual and the observed world? And what would it mean to share it?

If we assume that $\varepsilon_Y$ captures some latent person-specific health attributes (weight, age, BMI, etc.) then it would certainly make sense to share it if our intent was to predict the counterfactual headache duration for *the same person*. Alternatively, the variable could potentially represent the effect of environmental factors, or even the randomness in the actual amount of the active substance in the aspirin tablet of a given dose, due to tolerances in the manufacturing process. In that case, it might be counterintuitive to share the noise variable. If, in the hypothetical setting, we gave our friend a tablet with an advertised dose, there is little reason to expect that the variation in active substance content would be the same as for the lower dose tablet. See Appendix M for examples of other machine learning-specific reasons why sharing all the noise variables might not be the right thing to do.

All that is to say: whether to share all sources of randomness between the observed and counterfactual settings can be contextual depending on the counterfactual problem at hand. However, the way counterfactuals are typically defined in the causal literature, all sources of randomness must be shared (Peters et al., 2017; Pearl, 2009; Pearl & Mackenzie, 2018) as we'll see in Section 3.2. Of course, we could still tackle counterfactual problems in which we share all sources of randomness in the probabilistic framework (Appendix N). Nonetheless, sharing all sources of randomness requires us to make much stronger assumptions (see Appendix O). Hence, a framework that requires us to make such assumptions when we don't need to for the actual question at hand is overly restrictive.

### 3.2. "Classical" Approach: Structural Causal Models

In this section, we will again show how the probabilistic approach relates to the standard tools in causal inference — specifically, the *Structural Causal Model*.

We think clarifying these connections is needed, as a surface view of the literature might make one confused about what tools are sufficient or adequate for answering counterfactual questions. For example, Peters et al. (2017) write that *"causal graphical models are not rich enough to predict counterfactuals"*. Instead, they claim that a class of models called Structural Causal Models (SCMs) is neces-

sary: *"Formally, SCMs contain strictly more information than their corresponding graph and law (e.g., counterfactual statements)"*. On the other hand, Kusner et al. (2017), use Causal Bayesian Networks to reason about questions of counterfactual nature. Others, yet, make statements implying that standard probability theory is also insufficient: *"Counterfactuals, however, cannot be expressed through probabilistic conditioning alone."* (Tavares et al., 2021).

Informally, a Structural Causal Model (SCM) is a Bayesian Network on random variables $\mathbf{X}$ in which each variable $X_k$ is a deterministic function of its parents and a latent noise variable $N_j$. Formally, we can define SCMs as follows (Peters et al., 2017, §6.2):

**Definition 3.1. Structural Causal Model (SCM).** A structural causal model $\mathfrak{C}$ on a set of random variables $\mathbf{X}$ consists of a tuple $\langle \mathcal{S}, \mathcal{G}, p_{\mathbf{N}} \rangle$ where **1)** $\mathcal{G}$ is a directed acyclic graph on $\mathbf{X}$, **2)** $\mathcal{S}$ is a collection of $d$ structural assignments:

$$X_j := f_j\left(\mathbf{X}_{\mathbf{PA}_j^{\mathcal{G}}}, N_j\right), \quad j = 1, \ldots, d$$

in which $\mathbf{X}_{\mathbf{PA}_j^{\mathcal{G}}} \subseteq \{X_1, \ldots, X_d\} \setminus \{X_j\}$ are the parents of $X_j$ in graph $\mathcal{G}$ and $N_j$ are the latent noise variables, and **3)** $p_{\mathbf{N}}$ is a probability distribution over the noise variables $\{N_1, \ldots, N_d\}$ which are assumed to be independent, i.e. $p_{\mathbf{N}}(\mathbf{n}) = \prod_{j=1}^d p_{N_j}(n_j)$. The SCM uniquely specifies the distribution of the variables $\mathbf{X} \cup \mathbf{N}$.

A Structural Causal Model implies a Bayesian Network on $\mathbf{X}$ with the same graph $\mathcal{G}$. This is because each functional assignment $X_j := f_j\left(\mathbf{X}_{\mathbf{PA}_j^{\mathcal{G}}}, N_j\right)$ and the corresponding distribution function over the noise variable $N_j$ yield a conditional distribution on $X_j$ given its parents:

$$p(x_j|\mathbf{x}_{\mathbf{PA}_j^{\mathcal{G}}}) = \int \delta\left(x_j - f_j(\mathbf{x}_{\mathbf{PA}_j^{\mathcal{G}}}, n_j)\right) p(n_j) dn_j$$

Since the functional assignments entail the same set of conditional independencies as a Bayesian Network with a graph $\mathcal{G}$, the SCM also yields the same factorisation of the joint $p(\mathbf{x}) = \prod_j p(x_j|\mathbf{x}_{\mathbf{PA}_j^{\mathcal{G}}})$. In that sense, an SCM can be viewed as a Bayesian Network on $\mathbf{X}$ with additional structure; it conveys more than a Bayesian Network on $\mathbf{X}$. From another point of view, an SCM can be viewed as a restricted class of Bayesian Networks on $\mathbf{X} \cup \mathbf{N}$.

In that framework, to specify the counterfactual distribution, one can use the following construction:

**Definition 3.2. Counterfactuals in Structural Causal Models.** Given an SCM $\mathfrak{C} = \langle \mathcal{S}, \mathcal{G}, p_{\mathbf{N}} \rangle$ and an observation of the variables $\mathbf{X} = \mathbf{x}$, we can define a *counterfactual SCM* $\mathfrak{C}_{CF}$ on $\mathbf{X}^*$ by replacing the distribution of the noise variables:

$$\mathfrak{C}_{CF} = \langle \mathcal{S}, \mathcal{G}, p_{\mathbf{N}^*} \rangle$$

The altered distribution on the noise variables is given by $p_{\mathbf{N}^*}(\mathbf{n}) = p_{\mathbf{N}}(\mathbf{n}|\mathbf{X}=\mathbf{x})$. In other words, the posterior over the noise variables given $\mathbf{X}=\mathbf{x}$ in the original SCM $\mathfrak{C}$ becomes the prior for the new counterfactual SCM $\mathfrak{C}_{CF}$. The noise variables in this new counterfactual SCM need not be independent anymore.

Counterfactual statements can then be computed by intervening (in the sense of the rule given in def. 2.1) on the Bayesian Network on $\mathbf{X}^*$ with graph $\mathcal{G}$ implied by the counterfactual SCM $\mathfrak{C}_{CF}$.

This definition results in an inference procedure that is no different than what we've been doing in a joint model over both settings so far (Appendices N and N.1). Following this definition, we would first compute the posterior over the unobserved variables $\mathbf{N}$ given the observed data, and then run inference using that posterior as a prior in a second *counterfactual* model (which might have been altered to represent that we've intervened on the generative process). This is equivalent to assuming a joint model over both settings in which $\mathbf{N}$ are shared, and the remaining groups of variables $(\mathbf{X}, \mathbf{X}^*)$ are independent conditioned on $\mathbf{N}$. Pearl would likely agree with that point, seeing as he himself proposed such a 'twin network' model as a way to do Bayesian inference in SCMs (Balke & Pearl, 1994; Pearl, 2009, §7.1.4). Hence, we can use SCMs and the counterfactual operator in Def. 3.2 to define our assumptions over all settings of interest, and it *is* just a special case of the probabilistic modelling approach. As such, Structural Causal Models are sufficient for counterfactual inference, but not necessary. Furthermore, the need to specify everything in terms of 'structural' relations can be overly stringent for answering some types of counterfactual questions, as we illustrated in Section 3.1.

## 4. Discussion

In the previous sections, we illustrated, as concretely as possible, how to answer causal inference questions within the probabilistic modelling framework. In this section, we will argue why this perspective matters, why it is useful, and why there is a prevalence of statements that on the surface appear to oppose the main claim of this paper.

### 4.1. Benefits of a Probabilistic Framework

The probabilistic modelling framework is very flexible when dealing with settings that deviate from those foreseen by pre-existing frameworks, such as the Causal Bayesian Networks framework. In such cases, it's often a great layer of abstraction to fall back on, or formulate new frameworks and tools in. **(1)** For example, Von Kügelgen et al. (2023) describe a different type of causal question they term "backtracking counterfactuals". The authors explicitly define a "backtracking counterfactual" with a joint distribution over

the observed and the counterfactual settings. **(2)** Another example is modelling causal questions using tools other than graphical models. It is well-known, e.g. in the probabilistic programming literature (Milch et al., 2007), that a wide range of interesting models can't be represented with a Bayesian Network (or any 'causal' variant thereof): *"Despite widespread use, causal graphs cannot easily express many real-world phenomena"* (see Appendix P or Park et al. (2023) for examples). When taking the probabilistic modelling approach, tackling interventional and counterfactual questions in a probabilistic programming framework is conceptually no different than with a BN. The key principle is to specify all assumptions over all the tasks or settings of interest. Whether this is done through a probabilistic programming language, a BN, or otherwise, is secondary. **(3)** Yet another example is doing causal inference when we have data available from both the interventional and the observational settings (Colnet et al., 2023). Again, in this case, specifying a joint model over the two settings makes it obvious how to condition on data from the two sources to infer the shared model parameters. Although this is possible in the causal graphical modelling framework, it's arguably embarrassingly *natural* in the probabilistic one.

Secondly, with the probabilistic modelling approach, we can specify the joint and make inferences in the model however we want. In, e.g., the SCM framework, we are prescribed to follow the "abduction, action, prediction" steps. The cumbersomeness of this added rigidity is illustrated in Appendix N when computing counterfactuals for Example 2.

Lastly, the probabilistic framework can lower the barrier to entry to causal modelling to **1)** many in the machine learning community already familiar with that framework, and **2)** those that would have otherwise been turned off by having to learn a narrowly specialised tool.

In our view, there is plenty of potential for useful work at the intersection of causality and probabilistic modelling. In recent history, that has certainly been the case. For instance, Mossé et al. (2024) formally show that causal queries are always reducible to purely probabilistic queries to prove that causal and probabilistic languages have equal computational complexity. The movement to study identifiability (see Appendix J.1) in deep learning broadly has been heavily propelled and influenced by the causal and Independent Component Analysis (ICA) communities (see, for example, the plethora of references to causality and ICA in (Locatello et al., 2019)). For instance, Von Kügelgen et al. (2021) study identifiability in a strictly probabilistic model (Von Kügelgen et al. (2021), Figure 1)), but are evidently inspired by identifiability concerns in causality, and draw connections between their probabilistic model and interventions in a causal one. This is also the case in Locatello et al. (2020), who even explicitly write down the joint distribution over all

the variables of interest as recommended in this paper. Vice-versa, the works on nonlinear ICA (Hyvarinen et al., 2019; Hyvärinen et al., 2023) have led to plenty of novel results on identifiability of causal representation learning (Hyvärinen et al., 2024; von Kügelgen et al., 2023). Understanding that causal tools can be seen as syntactic shorthands for specifying joint distributions is, in our view, productive towards these endeavours.

## 5. Alternative Views

We wouldn't write this work if our main claim — that causal inference problems can be tackled with standard probability and statistics — wasn't seemingly frequently contradicted in the literature. One does not have to look far to find claims by Pearl that "we need to enrich our language with a *do*-operator" (Pearl, 2019a), and a plethora of machine learning papers mirror this sentiment (Appendix D).

**Causal-statistical dichotomy** Is the answer to the disagreement on the distinctness of causal and statistical approaches to inference clear-cut? Or can it be considered a matter of semantics or a difference in terminology? We thoroughly explore this question in Appendix Q; in short, it is our belief that the disagreement is primarily semantic. Nonetheless, we argue that the terminology that leads to the claim that 'causal and statistical inference are distinct' is deeply misleading, confuses newcomers to the field, and that the claims of a causal-statistical dichotomy should be done away with or carefully contextualised.

Statements about the limitations of the statistical framework are often directed at the caveats of using the "naïve" approach described in Section 2.2, and are used to stress the difficulty of specifying assumptions (Appendix F) and the importance of identifiability concerns (Appendix J.1) in the causal setting. However, attributing an inability to handle such caveats to the entirety of the statistical framework requires a historically inaccurate and reductionist characterisation of the field of statistics. Statistics research is well-accustomed to **1)** the necessity of subjective assumptions for inference, and **2)** tackling inference in non-static settings (e.g. domain shift, domain adaptation or off-policy reinforcement learning settings to name a few); two cornerstone attributes of the causal setting that Pearl uses as a key motivation for insisting on a causal-statistical dichotomy. In later works (Pearl et al., 2009), Pearl tempers the terminological distinction to that between *associational* and causal concepts instead. Although this delineation does not rely on unfaithfully limiting the scope of statistics — and is in our view clearer — it does impose the causal trademark on a broad class of concepts. Furthermore, the legacy of Pearl's activism for the causal-*statistical* distinction is still reflected in the language in present machine learning

literature (Appendix D includes such examples).

## 6. Conclusion

In this paper, we demonstrated that you *can* do causal inference through probabilistic modelling by defining a model over all settings of interest. The resulting approach was shown to be clear and general, and we argued for its utility for causality research, practice, and adoption. We gave concrete examples of causal inference questions on which we illustrated this approach, hopefully resolving the confusion surrounding whether you can do causal inference without the causal framework. We introduced various frameworks such as *Causal Graphical Models*, *Structural Causal Models* and the *do*-calculus, discussed their properties and proposed utility, and related them to the probabilistic modelling methodology. This can hopefully put to rest the myth that causal tools exist beyond the realm of probabilistic modelling, and lead to a more fruitful future collaboration across the communities.

To conclude, if there is 'one key takeaway' that we would like the reader to come away with, we want to expand upon MacKay's rule of writing down the probability of everything. In particular, we firmly recommend:

*Always write down the joint probability over all the settings you are interested in.*

## Acknowledgements

Special thanks must be reserved for Akil Hashmi and Matthew Ashman for their detailed feedback on the early drafts of this work. We also thank Melanie Pradier, Will Tebbutt, Taco Cohen, Johann Brehmer, Pim de Haan for helpful discussions and comments. Richard E. Turner is supported by the EPSRC Probabilistic AI Hub (EP/Y028783/1).

## Impact Statement

This paper argues for a conceptual shift in how causal inference is approached within the machine learning community, advocating for the sufficiency of the probabilistic modelling framework. The primary impact of this work is therefore on the community of researchers and practitioners. By framing causal problems within the familiar language of probability, this position has the potential to make causal reasoning problems more tractable and accessible, encouraging broader adoption and progress.

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

# A. Background and Definitions

## A.1. Log-normal Distribution

In the example, we will make use of the log-normal distribution, which is defined as follows:

**Definition A.1. Log-normal Distribution.** A random variable $X$ is distributed according to a log-normal distribution with parameters $(\mu, \sigma^2)$ when its logarithm is distributed according to a Gaussian distribution. In other words:

$$X \sim \log\mathcal{N}\left(\mu, \sigma^2\right) \qquad \textit{iff} \qquad \log X \sim \mathcal{N}\left(\mu, \sigma^2\right)$$

**Definition A.2. Multivariate Log-normal Distribution.** Similarly, a random vector $X$ on $\mathbb{R}^d$ is distributed according to a multivariate log-normal distribution with parameters $(\boldsymbol{\mu}, \mathbf{S}igma)$ when its elementwise logarithm is distributed according to a multivariate Gaussian distribution:

$$X \sim \log\mathcal{N}\left(\boldsymbol{\mu}, \boldsymbol{\Sigma}\right) \qquad \textit{iff} \qquad \begin{bmatrix} \log X_1 \\ \vdots \\ \log X_d \end{bmatrix} \sim \mathcal{N}\left(\boldsymbol{\mu}, \boldsymbol{\Sigma}\right)$$

A notable property is that product of two log-normal variables $X_1 \sim \log\mathcal{N}(\mu_1, \sigma_1^2)$ and $X_1 \sim \log\mathcal{N}(\mu_2, \sigma_2^2)$ is log-normal with parameters $(\mu_1 + \mu_2, \sigma_1^2 + \sigma_2^2)$, and a log-normal variable $X \sim \log\mathcal{N}(\mu, \sigma^2)$ raised to a constant power $c$ is also log-normal with parameters $(\mu, c^2\sigma^2)$.

## A.2. Bayesian Networks

Formally, a Bayesian Network can be defined as:

**Definition A.3. Bayesian Network (BN).** Random variables $\mathbf{X} = (X_1, \ldots, X_d)$ are a Bayesian Network with respect to a directed acyclic graph $\mathcal{G}$ if the random variables satisfy the following set of independencies[7]:

$$X_i \perp\!\!\!\perp \mathbf{X}_{\mathbf{ND}_i^\mathcal{G}} | \mathbf{X}_{\mathbf{PA}_i^\mathcal{G}}$$

where $\mathbf{X}_{\mathbf{ND}_i^\mathcal{G}}$ are the non-descendants of $X_i$ in $\mathcal{G}$, and $\mathbf{X}_{\mathbf{PA}_i^\mathcal{G}}$ are its parents. In other words, the non-descendants of $X_i$ are independent of $X_i$ given its parents.

The graph $\mathcal{G}$ in a Bayesian Network simply describes the set of conditional independencies between variables in $\mathbf{X}$. This set of conditional independencies leads to a simplified factorisation of the joint distribution:

$$p\left(x_1, \ldots, x_d\right) = \prod_{i=1}^d p\left(x_i | \mathbf{x}_{\mathbf{PA}_i^\mathcal{G}}\right) = \prod_{i=1}^d g_i\left(x_i, \mathbf{x}_{\mathbf{PA}_i^\mathcal{G}}\right) \tag{7}$$

where $g_i(x_i, x_{\mathbf{PA}_i^\mathcal{G}}) = p(x_i | x_{\mathbf{PA}_i^\mathcal{G}})$ is the conditional probability density function of $x_i$ given its parents[8]. This factorisation and the set of conditional independencies are two equivalent ways of defining a Bayesian Network; one necessarily implies the other.

## A.3. The rules of the *do*-calculus

In this section we give the rules of the *do*-calculus for reader's reference. It is helpful to define some extra notation before doing so. We'll write $\mathcal{G}_{\overline{\mathbf{X}}}$ for the graph obtained from $\mathcal{G}$ by deleting all the incoming edges for the nodes in set $\mathbf{X}$. Similarly, we'll write $\mathcal{G}_{\underline{\mathbf{X}}}$ for the graph obtained from $\mathcal{G}$ by deleting all the outgoing edges from the nodes in set $\mathbf{X}$.

For a Causal Graphical Model (or a Structural Causal Model) with a graph $\mathcal{G}$ and any disjoint subsets of variables $X, Y, Z$ and $W$, the rules of the *do*-calculus are as follows:

---

[7]When the random variables satisfy this condition, it is often said said that they satisfy the *Markov property* with respect to the graph $\mathcal{G}$.

[8]In the above definition of Bayesian Networks we considered the case when each node corresponds to one random variable. The definition can be trivially extended to the case in which we multiple random-variables correspond to each node. This could, for instance, occur when dealing with image data.

1. **Insertion/deletion of observations**

$$p(\mathbf{y}|\mathbf{z}, \mathbf{w}, do(\mathbf{X} = \mathbf{x})) = p(\mathbf{y}|\mathbf{w}, do(\mathbf{X} = \mathbf{x}))$$

if $\mathbf{Y}$ and $\mathbf{Z}$ are $d$-separated by $\mathbf{X}, \mathbf{W}$ in $\mathcal{G}_{\overline{\mathbf{X}}}$

2. **Action/observation exchange**

$$p(\mathbf{y}|\mathbf{w}, do(\mathbf{X} := \mathbf{x}, \mathbf{Z} = \mathbf{z})) = p(\mathbf{y}|\mathbf{z}, \mathbf{w}, do(\mathbf{X} := \mathbf{x}))$$

if $\mathbf{Y}$ and $\mathbf{Z}$ are $d$-separated by $\mathbf{X}, \mathbf{W}$ in $\mathcal{G}_{\overline{\mathbf{X}}\underline{\mathbf{Z}}}$.

3. **Insertion/deletion of actions**

$$p(\mathbf{y}|\mathbf{w}, do(\mathbf{X} := \mathbf{x}, \mathbf{Z} = \mathbf{z})) = p(\mathbf{y}|\mathbf{w}, do(\mathbf{X} := \mathbf{x}))$$

if $\mathbf{Y}$ and $\mathbf{Z}$ are $d$-separated by $\mathbf{X}, \mathbf{W}$ in $\mathcal{G}_{\overline{\mathbf{X}}\,\overline{\mathbf{Z}}_{(\mathbf{w})}}$ where $\mathbf{Z}_{(\mathbf{W})}$ is the subset of nodes in $\mathbf{Z}$ that are not ancestors of any nodes in $\mathbf{W}$ in graph $\mathcal{G}_{\overline{\mathbf{X}}}$.

## B. Example Derivations

### B.1. Joint log-normal distribution derivation for the aspirin example

To show that $[Z, T, Y]^\top$ are jointly log-normal as depicted in eq. 8, we can use the definition in eq. 1 to write:

$$\log \begin{bmatrix} Z \\ T \\ Y \end{bmatrix} = \begin{bmatrix} \log Z \\ a \log Z + \log \varepsilon_T \\ (b - ac) \log Z - c \log \varepsilon_T + \log \varepsilon_Y \end{bmatrix} = \underbrace{\begin{bmatrix} 1 & 0 & 0 \\ a & 1 & 0 \\ (b - ac) & (-c) & 1 \end{bmatrix}}_{A} \begin{bmatrix} \log Z \\ \log \varepsilon_T \\ \log \varepsilon_Y \end{bmatrix}$$

Since $\log[Z, \varepsilon_T, \varepsilon_Y]^\top$ are jointly normal distributed as $\mathcal{N}(\boldsymbol{\mu}, \boldsymbol{\Sigma})$, where $\boldsymbol{\mu} = [\mu_Z, 0, 0]^\top$ and $\boldsymbol{\Sigma} = \text{diag}([\sigma_Z^2, \sigma_T^2, \sigma_y^2]^\top)$, an affine transformation of these variables is also a Gaussian with mean $A\boldsymbol{\mu}$ and covariance $A\Sigma A^\top$. Hence, $[Z, T, Y]^\top$ are jointly distributed as $\log \mathcal{N}(A\boldsymbol{\mu}, A\Sigma A^\top)$. Evaluating the matrix multiplications yields the expression in eq. 8:

$$\begin{bmatrix} Z \\ T \\ Y \end{bmatrix} \sim \log \mathcal{N} \left( \begin{bmatrix} \mu_Z \\ a\mu_Z \\ (b - ac)\mu_Z \end{bmatrix}, \begin{bmatrix} \sigma_Z^2 & a\sigma_Z^2 & (b-ac)\sigma_Z^2 \\ a\sigma_Z^2 & a^2\sigma_Z^2+\sigma_T^2 & a(b-ac)\sigma_Z^2-c\sigma_T^2 \\ (b-ac)\sigma_Z^2 & a(b-ac)\sigma_Z^2-c\sigma_T^2 & (b-ac)^2\sigma_Z^2+c^2\sigma_T^2+\sigma_Y^2 \end{bmatrix} \right) \tag{8}$$

### B.2. Expectation of $Y$ conditioned on $T$ in the aspirin example

To obtain the expression in eq. 9, first denote the mean and the covariance matrix for the joint on $\log[Z, T, Y]^\top$ as $\overline{\boldsymbol{\mu}}$ and $\overline{\Sigma}$, i.e. $\log[Z, T, Y]^\top \sim \mathcal{N}(\overline{\boldsymbol{\mu}}, \overline{\Sigma})$. Using the marginalisation and conditioning properties of the multivariate normal (Wikipedia contributors, 2020) we get:

$$\log \begin{bmatrix} T \\ Y \end{bmatrix} \sim \mathcal{N} \left( \begin{bmatrix} \overline{\mu}_2 \\ \overline{\mu}_3 \end{bmatrix}, \begin{bmatrix} \overline{\Sigma}_{22} & \overline{\Sigma}_{23} \\ \overline{\Sigma}_{32} & \overline{\Sigma}_{33} \end{bmatrix} \right)$$

$$\log Y | \log T \sim \mathcal{N} \Big( \underbrace{\overline{\mu}_3 + \overline{\Sigma}_{32}\overline{\Sigma}_{22}^{-1}(\log T - \overline{\mu}_2)}_{\mu_{Y|T}}, \underbrace{\overline{\Sigma}_{33} - \overline{\Sigma}_{32}\overline{\Sigma}_{22}^{-1}\overline{\Sigma}_{23}}_{\sigma_{Y|T}^2} \Big)$$

Again, since $\log Y | \log T$ is Gaussian, $Y|T$ is log-normal distributed. We can then use the property that for a log-normal distribution $\log \mathcal{N}(\mu, \sigma^2)$ the expected value is $\exp(\mu + 0.5\sigma^2)$. Plugging in the values for $\overline{\boldsymbol{\mu}}$ and $\overline{\Sigma}$ into equations for $\mu_{Y|T}$ and $\sigma_{Y|T}^2$ then yields the expression in Equation (9).

$$\mathbb{E}[Y|T] = \exp \left( \frac{a(b-c)\sigma_Z^2 - c\sigma_T^2}{\sigma_Z^2 + c^2\sigma_T^2 + \sigma_Y^2}(T - a\mu_z) + \underbrace{\text{const.}}_{\substack{\text{independent} \\ \text{of } T}} \right) \tag{9}$$

## B.3. Derivation of the joint distribution in the interventional aspirin example

We can derive the expression for the joint distribution in Equation (2) as follows:

$$p(z^*, t^*, y^*, \boldsymbol{\theta}, \mathcal{D}) = \tag{10}$$

$$= p(z^*, t^*, y^* | \boldsymbol{\theta}, \cancel{\mathcal{D}}) p(\mathcal{D} | \boldsymbol{\theta}) p(\boldsymbol{\theta}) \tag{11}$$

$$= q_{\boldsymbol{\theta}}(z^*, t^*, y^*) \left( \prod_{i=1}^{N} p_{\boldsymbol{\theta}}(z_i, t_i, y_i) \right) p(\boldsymbol{\theta}) \tag{12}$$

$$= q_{\boldsymbol{\theta}}(z^*) q_{\boldsymbol{\theta}}(t^* | \cancel{z^*}) q_{\boldsymbol{\theta}}(y^* | z^*, t^*) \left( \prod_{i=1}^{N} p_{\boldsymbol{\theta}}(z_i) p_{\boldsymbol{\theta}}(t_i | z_i) p_{\boldsymbol{\theta}}(y_i | z_i, t_i) \right) p(\boldsymbol{\theta}) \tag{13}$$

$$= \underbrace{q_{\boldsymbol{\theta}}(z^*) q_{\boldsymbol{\theta}}(t^*) q_{\boldsymbol{\theta}}(y^* | z^*, t^*)}_{\text{Interventional world}} \underbrace{\left( \prod_{i=1}^{N} p_{\boldsymbol{\theta}}(z_i) p_{\boldsymbol{\theta}}(t_i | z_i) p_{\boldsymbol{\theta}}(y_i | z_i, t_i) \right) p(\boldsymbol{\theta})}_{\text{Observed world}} \tag{14}$$

Let's go through and make explicit the assumptions we exploited in the above derivation.

Firstly, in lines 11 and 12 we used the independence assumption given by the graphical model that all the observed examples $(Z_i, T_i, Y_i)$ and the interventional outcome $(Z^*, T^*, Y^*)$ are independent given the model parameters $\boldsymbol{\theta}$. I.e. once we know the parameters, the duration of one person's headache, its severity, and the dose they took is independent of how much aspirin someone else had taken, how long their headache lasted for, and its severity.

Secondly, in line 13 we used the fact that the aspirin dose $T^*$ in the interventional setting is independent of the headache severity $Z^*$, hence $q_{\boldsymbol{\theta}}(t^* | z^*) = q_{\boldsymbol{\theta}}(t^*)$. Let's consider in detail why this assumption makes sense. We're interested in a hypothetical intervention where we *assign* people a higher dose. To isolate the effect of aspirin, we are enquiring about an assignment of a given dose regardless of the subject's initial headache severity. Hence, the dose $T^*$ in the interventional setting ought to be independent of the headache severity $Z^*$.

These are all the assumptions we used to get us to eq. 14. We can plug in the exact forms of the distributions $q_{\boldsymbol{\theta}}(z^*)$, $q_{\boldsymbol{\theta}}(t^*)$, $q_{\boldsymbol{\theta}}(y^* | z^*, t^*)$ to arrive at Equation (2). Namely, by using the assumptions:

$$q_{\boldsymbol{\theta}}(y | t, z) = p_{\boldsymbol{\theta}}(y | t, z) \tag{15}$$
$$q(t) = \delta(t^* - t) \tag{16}$$
$$q_{\boldsymbol{\theta}}(z) = p_{\boldsymbol{\theta}}(z) \tag{17}$$

where $\delta(\cdot)$ is a Dirac delta function. Setting $T^*$ to be delta distributed results in the property:

$$p(\cdot) = \int p(\cdot, T^* = t) \, dt = \int p(\cdot | T^* = t) \underbrace{p(T^* = t)}_{\delta(t^* - t)} \, dt = p(\cdot | T^* = t^*) \tag{18}$$

I.e. marginalising out $T^*$ is equivalent to conditioning on $T^* = t^*$. This identity will come in useful below.

We can get the full joint in terms of probability density function which we know:

$$p(z^*, t^*, y^*, \boldsymbol{\theta}, \mathcal{D}) = \underbrace{p_{\boldsymbol{\theta}}(z^*) q_{\boldsymbol{\theta}}(t^*) p_{\boldsymbol{\theta}}(y^* | z^*, t^*)}_{\text{Interventional world}} \underbrace{\left( \prod_{i=1}^{N} p_{\boldsymbol{\theta}}(z_i) p_{\boldsymbol{\theta}}(t_i | z_i) p_{\boldsymbol{\theta}}(y_i | z_i, t_i) \right) p(\boldsymbol{\theta})}_{\text{Observed world}} \tag{19}$$

## B.4. Conditional expectation of $Y^*$ in the intervention aspirin example

To derive the expression in Equation (3), we can write:

$$\mathbb{E}[Y^*|\mathcal{D}] = \int y^* p(y^*|\mathcal{D}) dy^* \tag{20}$$

$$= \int y^* p(y^*|t^*, \mathcal{D}) dy^* \qquad \text{(using eq. 18)} \tag{21}$$

$$= \iiint y^* p(y^*, z^*, \boldsymbol{\theta}|t^*, \mathcal{D}) dz^* dy^* d\boldsymbol{\theta} \tag{22}$$

$$= \iiint y^* p(y^*|z^*, t^*, \boldsymbol{\theta}, \cancel{\mathcal{D}}) p(z^*|\boldsymbol{\theta}, \cancel{t^*}, \cancel{\mathcal{D}}) p(\boldsymbol{\theta}|\cancel{t^*}, \mathcal{D}) dz^* dy^* d\boldsymbol{\theta} \tag{23}$$

$$= \iiint \underbrace{y^* q_{\boldsymbol{\theta}}(y^*|z^*, t^*) q_{\boldsymbol{\theta}}(z^*) dz^* dy^*}_{\substack{\text{Calculate expectation in the intervened}\\ \text{world conditioned on posterior over } \boldsymbol{\theta}}} \underbrace{p(\boldsymbol{\theta}|\mathcal{D}) d\boldsymbol{\theta}}_{\substack{\text{Infer model parameters}\\ \text{using observed data } \mathcal{D}}} \tag{24}$$

where the cancellations in eq. 23 follow from the conditional independencies in the model.

## B.5. Marginal on $Y^*$ in the intervention aspirin example

In the interventional part of the model, $Y^*|Z^*, T^*$ is distributed in the same way as $Y|Z, T$ — $Y^*|Z^*, T^* \sim \log \mathcal{N}\left(\frac{(Z^*)^b}{(t^*)^c}, 1\right)$. We can equivalently specify $Y^* = \frac{(Z^*)^b}{(t^*)^c} \varepsilon_Y^*$ where $\varepsilon_Y^* \sim \log \mathcal{N}(0, 1)$. Hence, we can write:

$$\log \begin{bmatrix} Z^* \\ Y^* \end{bmatrix} = \begin{bmatrix} \log Z^* \\ b \log Z^* + \log \varepsilon_Y^* - c \log t^* \end{bmatrix} = \underbrace{\begin{bmatrix} 1 & 0 \\ b & 1 \end{bmatrix}}_{B} \begin{bmatrix} \log Z \\ \log \varepsilon_Y^* \end{bmatrix} + \underbrace{\begin{bmatrix} 0 \\ -c \log t^* \end{bmatrix}}_{d}$$

Again, as this is an affine transformation of normal-distributed variables, $\log[Z^*, Y^*]^\top$ is Gaussian:

$$\log \begin{bmatrix} Z^* \\ Y \end{bmatrix} \sim \mathcal{N}\left( \begin{bmatrix} \mu_Z \\ b\mu_Z - c \log t^* \end{bmatrix}, \begin{bmatrix} \sigma_Z^2 & b\sigma_Z^2 \\ b\sigma_Z^2 & b^2\sigma_Z^2 + \sigma_Y^2 \end{bmatrix} \right)$$

Consequently, by marginalising out $Z^*$ we get $Y^* \sim \log \mathcal{N}(b\mu_Z - c \log t^*, b^2\sigma_Z^2 + \sigma_Y^2)$. Using the property that for a log-normal distribution $\log \mathcal{N}(\mu, \sigma^2)$ the expected value is $\exp(\mu + 0.5\sigma^2)$, we obtain:

$$\mathbb{E}[Y^*] = \exp\left( b\mu_Z - c \log t^* + \frac{b^2\sigma_Z^2 + \sigma_Y^2}{2} \right) = (t^*)^{-c} \exp\left( b\mu_Z + \frac{b^2\sigma_Z^2 + \sigma_Y^2}{2} \right)$$

## B.6. Derivation of the joint distribution in the counterfactual aspirin example

Based on the graphical model, we can again write down a full joint distribution over the settings of interest:

$$p(z, t, y, z, t^*, y^*, \boldsymbol{\theta}, \mathcal{D}) =$$
$$= p(z, t, y, t^*, y^*|\boldsymbol{\theta}, \cancel{\mathcal{D}}) p(\mathcal{D}|\boldsymbol{\theta}) p(\boldsymbol{\theta}) \tag{25}$$

$$= \Big( p(z|\boldsymbol{\theta}) p(t|z, \boldsymbol{\theta}) p(y|t, z, \boldsymbol{\theta}) p(t^*|\cancel{y}, \cancel{t}, \cancel{z}, \boldsymbol{\theta}) p(y^*|t^*, \cancel{y}, \cancel{t}, z, \boldsymbol{\theta}) \Big) p(\mathcal{D}|\boldsymbol{\theta}) p(\boldsymbol{\theta}) \tag{26}$$

$$= \Big( p_{\boldsymbol{\theta}}(z) p_{\boldsymbol{\theta}}(t|z) p_{\boldsymbol{\theta}}(y|t, z) q(t^*) q_{\boldsymbol{\theta}}(y^*|t^*, z) \Big) \left( \prod_{i=1}^N p_{\boldsymbol{\theta}}(z_i) p_{\boldsymbol{\theta}}(t_i|z_i) p_{\boldsymbol{\theta}}(y_i|z_i, t_i) \right) p(\boldsymbol{\theta}) \tag{27}$$

$$= \Big( \underbrace{p_{\boldsymbol{\theta}}(t|z) p_{\boldsymbol{\theta}}(y|t, z)}_{\text{Observed world}} \underbrace{p_{\boldsymbol{\theta}}(z)}_{\substack{\text{Shared}\\\text{in both}}} \underbrace{q(t^*) p_{\boldsymbol{\theta}}(y^*|t^*, z)}_{\text{Counterfactual world}} \Big) \Big( \underbrace{\prod_{i=1}^N p_{\boldsymbol{\theta}}(z_i) p_{\boldsymbol{\theta}}(t_i|z_i) p_{\boldsymbol{\theta}}(y_i|z_i, t_i)}_{\text{Dataset likelihood}} \Big) p(\boldsymbol{\theta}) \tag{28}$$

Let's again unpack the assumptions made at each line. We used the conditional independence assumptions embodied in the graphical model to cancel terms in lines 25 and 26. In line 28, we make the additional assumption that $q_{\boldsymbol{\theta}}(y|t, z) = p_{\boldsymbol{\theta}}(y|t, z)$; in other words, the distribution of the counterfactual headache duration $Y^*$ conditioned on $Z$ and $T^*$ is distributed in the same way as the observed world counterpart $Y$ conditioned on $Z$ and $T$; the effect of aspirin dose and initial headache severity on headache duration is the same in the counterfactual world as in the observed world.

Using the properties of log-normal distributions, it can be seen that, conditioned on $\boldsymbol{\theta}$, variables $[Z, T, Y, Y^*]^\top$ are jointly distributed according to:

$$
\log \mathcal{N}\left(
\begin{bmatrix}
\mu_Z \\
a\mu_Z \\
(b-ac)\mu_Z \\
b\mu_Z - c\log t^*
\end{bmatrix},
\begin{bmatrix}
\sigma_Z^2 & a\sigma_Z^2 & (b-ac)\sigma_Z^2 & b\sigma_Z^2 \\
a\sigma_Z^2 & a^2\sigma_Z^2+\sigma_T^2 & a(b-ac)\sigma_Z^2-c\sigma_T^2 & ab\sigma_Z^2 \\
(b-ac)\sigma_Z^2 & a(b-ac)\sigma_Z^2-c\sigma_T^2 & (b-ac)^2\sigma_Z^2+c^2\sigma_T^2+\sigma_Y^2 & b(b-ac)\sigma_Z^2 \\
b\sigma_Z^2 & ab\sigma_Z^2 & b(b-ac)\sigma_Z^2 & b^2\sigma_Z^2+\sigma_Y^2
\end{bmatrix}
\right)
\tag{29}
$$

Hence, once we have a good estimate of the parameters $\boldsymbol{\theta}$, we can use the above expression and the conditioning/marginalisation formulas for the log-normal distribution to answer inferential questions about the counterfactual setting.

### B.7. Derivation of the conditional distribution in the counterfactual aspirin example

We can derive the expression for the conditional distribution in Equation (6) as follows:

$$
p(y^*|t, y, \mathcal{D}) = \int p(y^*|t, y, \boldsymbol{\theta}, \cancel{\mathcal{D}})p(\boldsymbol{\theta}|t, y, \mathcal{D})d\boldsymbol{\theta}
\tag{30}
$$

$$
= \mathbb{E}_{\boldsymbol{\theta}\sim p(\boldsymbol{\theta}|\mathcal{D}, t, y)}\left[p_{\boldsymbol{\theta}}(y^*|t, y)\right]
\tag{31}
$$

$$
= \mathbb{E}_{\boldsymbol{\theta}\sim p(\boldsymbol{\theta}|\mathcal{D}, t, y)}\left[\int p_{\boldsymbol{\theta}}(y^*|t^*, z, \cancel{t}, \cancel{y})p(z|t, y, \boldsymbol{\theta})dz\right]
\tag{32}
$$

$$
= \mathbb{E}_{\boldsymbol{\theta}\sim p(\boldsymbol{\theta}|\mathcal{D}, t, y)}\left[\int p_{\boldsymbol{\theta}}(y^*|t^*, z)p_{\boldsymbol{\theta}}(z|t, y)dz\right]
\tag{33}
$$

where we again relied on conditional independence properties entailed by the Bayesian Network in Figure 6 to arrive at the final expression.

From Equation (31), it is clear that once we have a good estimate of the parameters $\boldsymbol{\theta}$ from our dataset $\mathcal{D}$, we can answer questions about the counterfactual headache duration if we can evaluate the conditional density $p_{\boldsymbol{\theta}}(y^*|t, y)$. This conditional density has an analytical form. We can obtain the joint on $[T, Y, Y^*]^\top$ from equation 29 by marginalising out $Z$. This would yield:

$$
\begin{bmatrix}
T \\
Y \\
Y^*
\end{bmatrix} \sim \log \mathcal{N}\left(
\begin{bmatrix}
a\mu_Z \\
d\mu_Z \\
b\mu_Z - c\log t^*
\end{bmatrix},
\begin{bmatrix}
a^2\sigma_Z^2+\sigma_T^2 & ad\sigma_Z^2-c\sigma_T^2 & ab\sigma_Z^2 \\
ad\sigma_Z^2-c\sigma_T^2 & d^2\sigma_Z^2+c^2\sigma_T^2+\sigma_Y^2 & bd\sigma_Z^2 \\
ab\sigma_Z^2 & bd\sigma_Z^2 & b^2\sigma_Z^2+\sigma_Y^2
\end{bmatrix}
\right)
\tag{34}
$$

It is then straightforward to obtain the expression for the conditional $p_{\boldsymbol{\theta}}(y^*|t, y)$ from the log-normal joint over $[T, Y, Y^*]^\top$. Using a MAP estimate of the parameters, we could then obtain an approximate analytical (albeit lengthy) expression the conditional density $p(y^*|t, y, \mathcal{D})$ that we were seeking to infer.

## C. Derivations

### C.1. Proof of the parent adjustment formula

In this appendix, we give a proof of the parent adjustment formula.

The formula says that, in the case of an intervention on $X_t$ with an empty set of new parents, adjusting using the parents of $X_t$ in the observed Bayesian Network graph $\mathcal{G}$ is sufficient, i.e.:

$$
q(x_y) = q(x_y|x_t) = \int p(x_y|x_t, \mathbf{x}_{\mathbf{PA}_t^{\mathcal{G}}})p(\mathbf{x}_{\mathbf{PA}_t^{\mathcal{G}}})d\mathbf{x}_{\mathbf{PA}_t^{\mathcal{G}}}
\tag{35}
$$

Where $q(\cdot)$ are the distribution functions in the intervened setting, and $p(\cdot)$ correspond to the observed setting. We can show

that the parent adjustment formula holds by expanding:

$$q(x_y|x_t) = \frac{q(x_y, x_t)}{q(x_t)} = \frac{q(x_y, x_t)}{g^*(x_t)}$$

$$= \int \left( \int \frac{q(\mathbf{x})}{g^*(x_t)} d\mathbf{x}_{other} \right) d\mathbf{x}_{\mathbf{PA}_t^{\mathcal{G}}}$$

where the inner integral over $\mathbf{x}_{other}$ is taken with respect to all nodes $\{1, \ldots, d\}$ other than $\{t, y\} \cup \mathbf{PA}_t^{\mathcal{G}}$. Expanding the joint $q(\mathbf{x})$:

$$= \int \left( \int \frac{\cancel{g^*(x_t)} \prod_{i \neq t} g\left(x_i, x_{\mathbf{PA}_i^{\mathcal{G}}}\right)}{\cancel{g^*(x_t)}} d\mathbf{x}_{other} \right) d\mathbf{x}_{\mathbf{PA}_t^{\mathcal{G}}}$$

$$= \int \left( \int \frac{g_t(x_t, \mathbf{x}_{\mathbf{PA}_t^{\mathcal{G}}}) \prod_{i \neq t} g\left(x_i, x_{\mathbf{PA}_i^{\mathcal{G}}}\right)}{g_t(x_t, \mathbf{x}_{\mathbf{PA}_t^{\mathcal{G}}})} d\mathbf{x}_{other} \right) d\mathbf{x}_{\mathbf{PA}_t^{\mathcal{G}}}$$

$$= \int \left( \int \underbrace{\prod_i g\left(x_i, x_{\mathbf{PA}_i^{\mathcal{G}}}\right)}_{p(\mathbf{x})} d\mathbf{x}_{other} \right) \frac{1}{g_t(x_t, \mathbf{x}_{\mathbf{PA}_t^{\mathcal{G}}})} d\mathbf{x}_{\mathbf{PA}_t^{\mathcal{G}}}$$

$$= \int \left( \int p(\mathbf{x}) d\mathbf{x}_{other} \right) \frac{1}{p(x_t|\mathbf{x}_{\mathbf{PA}_t^{\mathcal{G}}})} d\mathbf{x}_{\mathbf{PA}_t^{\mathcal{G}}}$$

$$= \int p(x_y, x_t, \mathbf{x}_{\mathbf{PA}_t^{\mathcal{G}}}) \frac{1}{p(x_t|\mathbf{x}_{\mathbf{PA}_t^{\mathcal{G}}})} d\mathbf{x}_{\mathbf{PA}_t^{\mathcal{G}}}$$

$$= \int p(x_y|x_t, \mathbf{x}_{\mathbf{PA}_t^{\mathcal{G}}}) p(x_t|\mathbf{x}_{\mathbf{PA}_t^{\mathcal{G}}}) p(\mathbf{x}_{\mathbf{PA}_t^{\mathcal{G}}}) \frac{1}{p(x_t|\mathbf{x}_{\mathbf{PA}_t^{\mathcal{G}}})} d\mathbf{x}_{\mathbf{PA}_t^{\mathcal{G}}}$$

$$= \int p(x_y|x_t, \mathbf{x}_{\mathbf{PA}_t^{\mathcal{G}}}) p(\mathbf{x}_{\mathbf{PA}_t^{\mathcal{G}}}) d\mathbf{x}_{\mathbf{PA}_t^{\mathcal{G}}}$$

where the last line is the same as the right-hand side of equation 35.

## D. Claims of insufficiency of machine learning and statistics for causal inference

In this section, we give examples of apparent claims of insufficiency of machine learning, statistics, probability and probabilistic modelling and inference for causal reasoning. The quotes are intended to showcase the *seeming* disagreement with the main premise of the paper: that you can answer interventional and counterfactual inference questions with probabilistic conditioning alone. We note that many of the quotes could be interpreted as voicing a more nuanced opinion; even in these cases — as we note in Appendix Q — we believe they can be (and often are) easily misconstrued by newcomers to the field. Hence, they still point to a need for greater clarity of language when discussing what probabilistic methods can and cannot do.

---

> *"The field of causality was born from the observation that **probability theory and statistics cannot encode the notion of causality**, and so we need additional mathematical tools to support the enhanced view of the world involving causality."*

Park et al. (2023, p. 1)

Yet, the assumptions and axioms of probability theory are not subsumed or replaced in (Park et al., 2023), but are instead complimented with additional axioms that build upon or add to the probabilistic foundations. Hence, the work of Park et al. (2023) can be viewed as building a theoretical framework at a higher level of abstraction. This is similar to how we argued that causal graphical models or the *do*-calculus can be viewed as built on top of probabilistic modelling, albeit the framework of Park et al. (2023) is significantly more general.

> *"Many questions in everyday life as well as in scientific inquiry are causal in nature: "How would the climate have changed if we'd had less emissions in the '80s?", "How fast could I run if I hadn't been smoking?", or "Will my headache be gone if I take that pill?". **None of those questions can be answered with statistical tools alone**, but require methods from causality to analyse interactions with our environment (interventions) and hypothetical alternate worlds (counterfactuals), going beyond joint, marginal, and conditional probabilities Peters et al. (2017)."*

> Pawlowski et al. (2020, p. 1)

In this work, we have shown that such questions can be answered using statistical tools (Bayesian Networks) alone. Like many of the quotes that will follow, their claims of insufficiency of "statistical tools" can likely be attributed to ascribing a very specific meaning to the term "statistical" (as discussed in detail in Appendix Q). Namely, if one takes it as a given that the goal of "statistics" is only to describe the data from the "observed distribution" well (something we argue is an unfair assessment of the field in Appendix Q), then the quote makes sense in a self-fulfilling manner. For instance, they clarify that "In addition, deep generative models have been heavily used for (unsupervised) representation learning with an emphasis on disentanglement. However, even when these methods faithfully capture the distribution of observed data, they are capable of fulfilling only the association rung of the ladder of causation" (Pawlowski et al., 2020, p. 5). In our view, it should come as no surprise that these models can only answer associational questions, when they are deployed to model associations amongst observed variables only. This, however, does hint at the restricted meaning the authors imbue the term "statistical" with in their work.

---

> *"Crucially, unlike conventional Bayesian networks, the conditional factors [in Structural Casual Models defined] above are imbued with a causal interpretation. This enables [Structural Causal Models] to be used to predict the effects of interventions [...]."*

> Pawlowski et al. (2020, p. 2)

We have successfully used "conventional" Bayesian Networks in this work to answer causal questions. Of course, the difference likely lies in interpreting "conventional" as "the way in which Bayesian Networks are conventionally deployed". If one considers it "conventional" to only specify Bayesian Networks over the variables in the observed world (rather than a hypothetical intervened-upon or counterfactual world), then the meaning of the quote is clear. We would, however, argue that we're not breaking any conventions by using Bayesian Networks to specify models in this way.

---

> *"Scientific inquiry is invariably motivated by causal questions: "how effective is X in preventing Y ?", or "what would have happened to Y had X been x?". Such questions cannot be answered using statistical tools alone."*

> Ribeiro et al. (2023, p. 1)

The quote again makes sense, but only under a very specific meaning attributed to the term "statistical" (Appendix Q).

---

> *"Counterfactuals, however, cannot be expressed through probabilistic conditioning alone."*

> Tavares et al. (2021, p. 1)

In Section 3, we answer counterfactual questions using conditioning in a probabilistic model. The intended meaning of the quote is likely that probabilistic conditioning of the random variables in the observed variables will, in general, not correspond to the intended question when doing interventional or counterfactual inference (see Section 2.1). However, with the specific wording the authors chose, it's easy to come away with a different message.

---

> *"While Bayesian networks can uncover statistical correlations between factors, SCMs can be used to answer higher-order questions of cause-and-effect, up in the ladder of causation."*

> Ke et al. (2020, p. 1)

Although the authors do not claim Bayesian Networks *cannot* be used to answer questions of cause-and-effect, it is easy to misconstrue the sentence as implying such a contrast in capabilities of SCMs and Bayesian Networks.

---

*"Many statisticians are reluctant to deal with problems involving causal considerations because **we lack the mathematical notation for distinguishing causal influence from statistical association**."*

Pearl (1995, p. 1)

In this work, we show it is perfectly adequate to notate both questions about statistical association and causal influence in the mathematical notation of probability theory. At the time, Rubin (1974) had already accomplished a similar feat. Pearl likely means that we lack *accessible* mathematical notation for answering causal questions, and it is clear he did not believe the notation of Rubin was up to the standard. We believe that the method and mathematical notation for answering causal questions presented in this work is both clear and accessible.

---

The following are yet a few more examples of statements that make sense under the reappropriated meaning of "statistical" discussed in Appendix Q:

*"Examples include interventional questions: "What if I make it happen?" and retrospective or explanatory questions: "What if I had acted differently?" or "What if my flight had not been late?". **Such questions cannot be articulated, let alone answered, by systems that operate in purely statistical mode**."*

Pearl (2019c, p. 1)

Once again, the quote makes sense if we interpret "statistical mode" to mean "observational" or "concerned with *i.i.d.* random variables in the observed setting only".

*"Traditional statistical learning techniques only allow us to answer questions that are inherently associative in nature."*

Rasal et al. (2022, p. 2)

*"Causal learning is motivated by shortcomings of statistical learning"*

Schölkopf and von Kügelgen (2022, p. 2)

The authors of the last quote later clarify that the cause of the issue is with wrongly making the *i.i.d.* assumption: *"a crucial aspect that is often ignored is that we assume [in statistical learning] that the data are i.i.d."* Schölkopf & von Kügelgen (2022). However, that assumption is not *intrinsic* to a statistical approach, it is just common.

---

*"Current statistical methods, in contrast, exploit associative relationships to improve prediction accuracy, which is sound as long as distributions do not shift [...]. But interventions in a system generally lead to distribution shift, and correlation does not imply causation."*

Paleyes et al. (2023, p. 2)

This quote could easily be interpreted as claiming that statistical methods are not able to correctly answer inference questions in the presence of distribution shift, in particular, that induced by interventions. This is clearly possible with statistical tools, such as Bayesian networks, which were "current" at the time of writing of the paper.

---

*"To model interventions requires the ability to represent context-specific independence: in the context of an intervention on a variable, any influences that normally have a causal effect on that variable are removed. **Bayesian networks and factor graphs lack the ability to represent such context-specific independence** and so are unable to represent interventions in sufficient detail to reason about conditional independence properties. Pearl's innovative do calculus was proposed as an additional mechanism outside of probabilistic inference which allows for reasoning about interventions and hence causality."*

Winn (2012, p. 1)

---

Certain quotes could be read as making claims about the necessity of the use of a particular framework, such as Pearl's

*causal models*:

> "**Answering the interventional and counterfactual queries requires** *modeling policy interventions* **using a causal model**."
>
> Hızlı et al. (2023, p. 1)

---

Many works do, however, seem to appropriately caveat the claims about the insufficiencies of machine learning and statistical methods by pointing these are *tendencies* in the research landscape:

> "*ML [Machine Learning] methods excel at fitting data and making predictions based on statistical associations, but are generally unable to answer causal questions.*"
>
> Kekić et al. (2022, p. 2)

## E. General "recipe" for tackling interventional problems with Bayesian Networks

Let's briefly summarise how we *could* approach interventional inference using Bayesian Networks by defining a joint model over the observed and the intervened-upon setting. This approach summarises the general steps that we used to model the effects of an intervention in the preceding aspirin example:

1. Specify a Bayesian Network over the observed world variables $\boldsymbol{X}$.

2. Define the joint distribution $p_{\boldsymbol{\theta}}(\boldsymbol{x})$ over the observed variables parameterised by some (usually unknown) parameters $\boldsymbol{\theta}$.

3. Specify a Bayesian Network over the corresponding variables $\boldsymbol{X}^*$ in the intervened-upon setting by considering which dependencies have been altered/removed compared to the observed setting.

4. Define the form of the joint distribution $q_{\boldsymbol{\theta}}(\boldsymbol{x}^*)$ in the intervened-upon setting, specifying which parts are to remain the same as in the observed setting, and giving the form for the remaining ones.

5. The complete joint model over the two settings is constructed from the two Bayesian Networks over $\mathbf{X}$ and $\mathbf{X}^*$ by drawing arrows from $\boldsymbol{\theta}$ to variables in both world, reflecting that the parameters $\boldsymbol{\theta}$ are assumed to be the same in both settings, and that $\mathbf{X}$, $\mathbf{X}^*$ are independent once the parameters are known.

In Example 1, we looked at inferring an interventional quantity of interest under one of many possible interventions. The intervention detailed above, in which the treatment (aspirin dose) is independent of the headache severity, allowed us to determine the efficacy of aspirin. We could have just as well extended to different, possibly probabilistic, interventions. For instance, what if we were wondering what would happen if the national department for health issued a new recommendation to increase aspirin dosage? In this interventional setting, the aspirin dose would still likely be dependent on the headache severity; however, as a result of the intervention, $p_{T^*|Z^*}(t^*|z^*)$ would differ from the observed world. We could specify a similar joint model as in Example 1 by designating how we believe the decision mechanism for choosing a dose based on the headache severity would change subject to the intervention.

Effects of such "soft" probabilistic interventions are of interest in many domains, especially with regards to policy-making. For example, consider predicting the effect of a regulation banning cigarette advertisement on the national life expectancy. We could construct an equivalent model with $T$ representing cigarette usage, $Y$ the life expectancy, and $Z$ the underlying health attitude. In this case, the regulation is likely to alter the conditional distribution of cigarette usage given someone's health attitude — $p_{T^*|Z^*}(t^*|z^*)$ — hopefully lowering the expected number of cigarettes consumed for any given $z^*$; nevertheless, $T^*$ would still be dependent on $Z^*$ after this intervention.

Furthermore, Bayesian Networks are only one of the many tools we could use to define a joint model over the interventional and observed settings; we could specify our assumptions in a variety of other equivalent ways. The key part is to recognise that the variables in the interventional and observational settings might be distributed differently, and then choose a method to define how the two settings are related by defining a joint distribution over $\mathbf{X}$, $\mathbf{X}^*$. In fact, as we discuss in section **??**, Bayesian Networks – and graphical models in general – are somewhat limited in that they cannot describe certain settings that, for instance, probabilistic programming languages can.

# F. Challenges of specifying an interventional model

In the aspirin example, it was fairly easy to see how to model the intervention; we could quite intuitively deduce which parts of the intervened-upon world to keep shared with the real world and which ones to alter. However, it is worth noting that in general there is significant nuance to it.

In a more general setting, consider having an outcome variable $Y$ (e.g. headache duration), a treatment variable $T$ (e.g. aspirin dose), and a set of remaining variables $\mathbf{Z}$ in the observed world. The initial instinct, based on the aspirin example, might be to factorise the joint distribution $p(\mathbf{z}, t, y) = p(\mathbf{z})p(t|\mathbf{z})p(y|t, \mathbf{z})$ and simply replace the 'treatment-determining' mechanism $p(t|\mathbf{z})$ with a new 'intervention' mechanism $q(t|\mathbf{z})$ to get the joint over the interventional variables. For instance, in the intervened-upon setting, we could specify $T$ to be independent of $Z$ — $q(t|\mathbf{z}) = \delta(t - t^*)$ — mirroring what we did in the aspirin example. Although this was adequate in said example, it is not necessarily so in the general case.

For instance, what if some of the variables in $\mathbf{Z}$ mediate the effect of the treatment $T$ rather than confound it? If $\mathbf{Z}$ contained a variable representing measurements of post-ingestion prostaglandin levels, for instance, making prostaglandin levels independent of the aspirin dose could "block" the main mechanism by which the aspirin affects the headache duration. This is illustrated in Figure 7a. When we want to infer the effects of an *action* to assign a given aspirin dose, we would imagine that in the intervened-upon world the physical effect of aspirin on the body is still the same as in the observed world; that is, we would believe that prostaglandin levels are still dependent (in the same way as in the observed setting) on the aspirin dose ingested[9].

Another important consideration is: what if there is another hidden confounder that we haven't accounted for? For instance, the subjects in the survey possibly based their aspirin dose decision on whether they had a fever; however, a fever might be indicative of the headache type (migraine, tension headache, etc.), which in turn affects the effectiveness of aspirin. We saw in section 2.1 how not accounting for a confounder could yield an inference outcome different from the one we intended. In the intervened setting, if we are asking about hypothetically administering someone a given dose regardless of their condition, we implicitly assume the dose assignment to be independent of the fever. However, if the model of the interventional setting doesn't reflect that assumption, as it doesn't account for some hidden confounder, the output of the inference procedure could be heavily misleading. In other words, there would be a mismatch between our interpretation of the output of the procedure, and what it actually corresponds to. In fact, this is one of the fundamental problems of causal inference: the hope that you have accounted for all the factors that could confound the relationship between the quantities of interest.

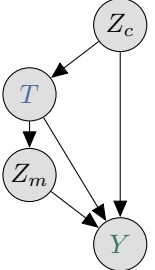

(a) Graphical model for an aspirin example with a mediator variable $Z_m$.

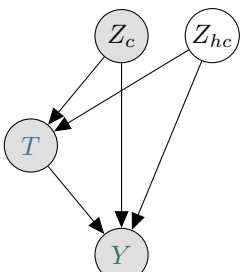

(b) Graphical model with an unobserved hidden confounder $Z_{hc}$.

*Figure 7.*

Judea Pearl's "The Book of Why" (Pearl & Mackenzie, 2018) and "Causality" (Pearl, 2009) do an excellent job showcasing how an incorrect treatment of such scenarios could lead to a misinterpretation of the outcome of an inference procedure. The takeaway message advocated by Pearl is that there is no universal way to generalise from the observed distribution to what we would interpret as an intervened-upon distribution; you need to make assumptions about the generative process for the data, and how an intervention would change it, to be able to do so.

---

[9]Note that in this example, which variables in $\mathbf{Z}$ mediate and which confound requires a very simple judgement call. Any measurements taken before the treatment decision was made are potential confounders, and any measurements taken after the treatment decision couldn't have possibly influenced the decision to take a given treatment. This is, however, complicated by the fact that there could be other unobserved variables that confound the treatment decision and the measurement taken after that decision was made.

# G. Interventional inference with machine learning models

The aspirin example presented in section 2.2 is a very simple three-variable task. We could derive analytical expressions for all conditional distributions of interest. In machine learning, however, we are often dealing with high-dimensional and highly non-linear data. For instance, in a medical setting we might be dealing with DNA sequences or CT images rather than something as simple as a scalar headache severity rating. How could we approach modelling interventions in these more complex settings?

We saw previously in eq. 5 that we can factorise the joint distribution in the intervened-upon world into conditional distribution functions shared with the observed world and those explicitly specified as part of the intervention. Each of the conditional distribution functions had the form $p_{\boldsymbol{\theta}}(\mathbf{x_k}|\mathbf{x_{PA}}_k^{\mathcal{G}})$. We can use a probabilistic model of choice to model these conditional distributions in a supervised-learning fashion using data from the observed world. For the variables without any parents[10], we could use any unsupervised learning method, e.g. Variational Autoencoders (VAEs) (Kingma & Welling, 2013) or Normalising Flows (Rezende & Mohamed, 2015), to model $g_{\boldsymbol{\theta}}(\mathbf{x_k}) = p_{\boldsymbol{\theta}}(\mathbf{x_k})$. A fairly general framework for specifying such systems using Gaussian Processes has been proposed by e.g. Silva & Gramacy (2010).

Running inference in these models can be very application-dependent. For instance, Louizos et al. (2017) consider a specific architecture based on VAEs for estimating Individual Treatment Effect for a specific graphical model with a partially-observable confounder. In the more general case, however, one can imagine using e.g. ancestral sampling to estimate any integrals of interest — first sample the variables that don't have any parents from $p_{\boldsymbol{\theta}}(\mathbf{x_k})$, then their children from $p_{\boldsymbol{\theta}}(\mathbf{x_k}|\mathbf{x_{PA}}_k^{\mathcal{G}})$ conditioned on the sampled values, their children's children and so on.

Hence, it should be clear that you *can* do causal inference using machine learning and deep learning methods. You also don't need to invoke any causal notation or causal-specific framework to do so.

# H. Markov Equivalence Classes

There can be multiple graphs that give the same set of conditional independencies in a Bayesian Network, and hence entail the same factorisation of the joint distribution. They are said to belong to the same *Markov Equivalence Class*. Recall that in the definition of a Bayesian Network, the graph was solely used to define said conditional independencies. Hence, all members of a Markov Equivalence Class imply the same restrictions on the joint distribution.

To give a concrete example of a Markov Equivalence Class, consider the graph in fig. 1 for the aspirin example. That graph does not entail any conditional independencies; the factorisation that we obtain from that Bayesian Network — $p(y, t, z) = p(y|t, z)p(t|z)p(z)$ — is valid for *any* probability distribution by the product rule of probability functions. I.e. the graph doesn't entail *any* conditional independence assumptions. Hence, any rearrangement of the edges that results in a Directed Acyclic Graph, as shown in Figure 8, yields the same set of conditional independencies.

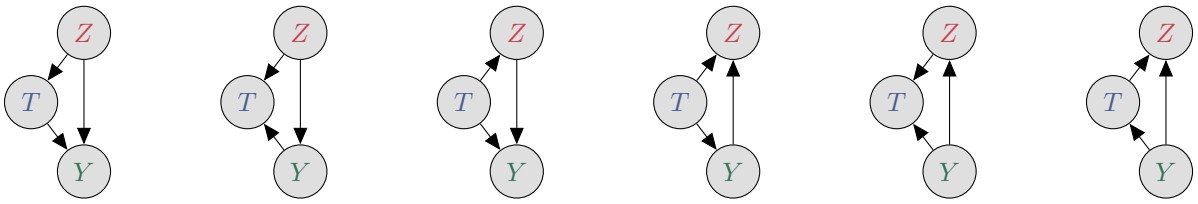

*Figure 8.* Bayesian Networks entailing the same set of conditional independencies (and consequently the same factorisation of the joint distribution).

For the purposes of fitting the observed distribution only, if we use the graph of a Bayesian Network as a constraint on the learnt distribution, it makes no difference which of the Bayesian Networks in the Markov equivalence class is being considered; they all imply the same restrictions (in terms of conditional independencies) on the observed distribution. If we tried to infer the graph by estimating the conditional independencies from observed data, we could only distinguish between different Markov Equivalence Classes, but not between the members of each class. Even in the limit of infinite data, there is no statistical test on the observed data to uniquely determine a member from within the equivalence class.

---

[10]Also called *exogenous*.

It should be clear that applying the rule in Def. 2.1 to the different graphs in Figure 8 will yield different interventional distributions. For instance, an atomic intervention on $T$ in the 6th graph in Figure 8 would make $T^*$ independent of $Z^*$ and $Y^*$ in the intervened-upon setting, whereas same intervention on $T$ for the 3rd graph wouldn't introduce any independencies.

As the different graphs will, however, yield a different "interventional" distribution through the application of the rule in Def. 2.1, this means that we can't uniquely determine the interventional distribution from the observed data by learning a graph. The interventional distribution is *non-identifiable* in the absence of further assumptions. These assumptions could, for example, come in the form of a particular graph (and hence, implicitly, a particular interventional distribution implied by it), or through constraints on the observed distribution (e.g. by assuming that the form of the conditional distribution of a child variable conditioned on its parents is a specific model class, such as an additive noise model (Peters et al., 2012)).

Note that this is a complete non-issue when using a joint model over all the settings of interest. By specifying a complete joint model over all the relevant tasks — like we did in the aspirin example — we are forced to explicitly make the assumptions about how these tasks are related. Hence, you can either choose to specify a single model over the observed setting while being mindful of the implicit assumptions and the Markov Equivalence Classes, or you can specify a joint model where the assumptions are crystal clear.

## I. Atomic interventions and Randomised Controlled Trials

The most prevalent type of an intervention is an *atomic intervention*. In an atomic intervention, no new edges are added in the modified model ($\mathbf{PA}_j^* = \emptyset$) and the random variable at node $j$ is set to a constant value $c_j$, i.e. $g^*(x_j) = \delta(x_j - c_j)$[11]. Intuitively, the modification to the graphical model compromises removing edges going into $j$ and forcing the variable $X_j$ to take on some fixed value. This corresponds exactly to the type of intervention we considered in the aspirin example. The post-intervention distribution for the case of an atomic intervention can be written as[12]:

$$q(\mathbf{x}) = \delta(x_j - c_j) \prod_{i \in \{1,\ldots,d\}\setminus\{j\}} g_i(x_i, \mathbf{x}_{\mathbf{PA}_i^{\mathcal{G}}})$$

For atomic interventions, and in fact for any intervention with an empty set of parents ($\mathbf{PA}_j^* = \emptyset$), the conditional distribution $q(\mathbf{x}_{-j}|x_j) = q(x_1, \ldots, x_{j-1}, x_{j+1}, \ldots, x_d|x_j)$ simplifies to:

$$q(\mathbf{x}_{-j}|x_j) = \left( g^*(x_j) \prod_{i \in \{1,\ldots,d\}\setminus\{j\}} g(x_i, \mathbf{x}_{\mathbf{PA}_i^{\mathcal{G}}}) \right) \frac{1}{g^*(x_j)} = \prod_{i \in \{1,\ldots,d\}\setminus\{j\}} g(x_i, \mathbf{x}_{\mathbf{PA}_i^{\mathcal{G}}}) \tag{36}$$

whenever $x_j$ is in the support of $g^*(x_j)$. I.e., the conditional distribution given $X_j = x_j$ is the same independently of the choice of $g^*(x_j)$, or whether the intervention is atomic or probabilistic.

### I.1. Interventions and randomised control trials

Linking interventional distributions obtained through this rule to experimental studies helps to establish some intuition for its workings. Specifically, atomic interventions have an intuitive relation to randomised control trials (RCTs).

In RCT studies, one randomly assigns the value (treatment) to a variable of interest $X_j$ irrespective of other factors. For instance, patients would be administered placebo or the real drug based solely on 'the flip of a coin'. This can of course be represented by an intervention in a Bayesian Network (assuming original BN represents a non-experimental setting), where the new set of parents of $X_j$ is an empty set, and its value is determined purely at random, according to some distribution $g^*(x_j)$ (e.g. $X_j \sim \text{Bern}(\frac{1}{2})$ if $X_j \in \{0, 1\}$ is picked based on a coinflip). As such, it can be seen that in the BN corresponding to the randomised control trial, the conditional distribution $p_{RCT}(\mathbf{x}_{-j}|x_j)$ is the same as the conditional distribution for a BN resulting from an atomic intervention fixing the value of $X_j$ to $x_j$:

$$p_{RCT}(\mathbf{x}_{-j}|x_j) = \frac{p_{RCT}(\mathbf{x})}{g^*(x_j)} = \left( g^*(x_j) \prod_{i \in \{1,\ldots,d\}\setminus\{j\}} g(x_i, \mathbf{x}_{\mathbf{PA}_i^{\mathcal{G}}}) \right) \frac{1}{g^*(x_j)} = \prod_{i \in \{1,\ldots,d\}\setminus\{j\}} g(x_i, \mathbf{x}_{\mathbf{PA}_i^{\mathcal{G}}})$$

---

[11]In the discrete case, a Kronecker delta function can equivalently be used.

[12]For the discrete case, when $\delta(x_j - c_j) \in \{0, 1\}$ is a Kronecker delta, this neatly simplifies to: $q(\mathbf{x}) = \prod_{i \in \{1,\ldots,d\}\setminus\{j\}} g(x_i, \mathbf{x}_{\mathbf{PA}_i^{\mathcal{G}}})$ if $x_j = c_j$ and 0 otherwise.

## J. Inference in interventional models and the *do*-calculus

In the aspirin example, in equations 20 - 3 we were able to obtain an expression for the conditional $q(y^*|t^*, \boldsymbol{\theta})$ in the interventional part of the model in terms of density functions that were the same between observed and interventional worlds: $q(y^*|t^*, \boldsymbol{\theta}) = \int p_{\boldsymbol{\theta}}(y^*|t^*, z^*)p_{\boldsymbol{\theta}}(z^*)dz^*$. Specifically, conditioning on $Z^*$ allowed us to do so. This is typically referred to as *adjusting* for $Z$.

The mathematics were simple for this three-variable model, however, what if the model was more complex? In Bayesian Networks, are there any shortcuts to expressing the conditional of interest in the intervened-upon world in terms of conditionals of the variables from the observed world only?

The definition for the intervention operation in BNs def. 2.1 gives an expression for the joint in term of conditionals from the observed setting and $g^*(x_j, \mathbf{x}_{\mathbf{PA}_j^*})$, which is specified as part of the intervention (eq. 5). From the joint, we can obtain any desired conditional of interest in the interventional setting by marginalising out; say, without loss of generality, we're interested in $q(x_1, \ldots, x_m | x_{m+1} \ldots x_n)$ for a BN on $d$ variables:

$$q(x_1, \ldots, x_m | x_{m+1} \ldots x_n) = \frac{q(x_1, \ldots, x_n)}{\int q(x_1, \ldots, x_n)d\mathbf{x}_{m+1:n}} = \frac{\int q(x_1, \ldots, x_d)d\mathbf{x}_{n+1:d}}{\int q(x_1, \ldots, x_d)d\mathbf{x}_{m+1:d}}$$

Where we can express $q(x_1, \ldots, x_d)$ in terms of densities from the observed distribution as desired. For complex models, such as neural networks, the fraction of the two integrals could be estimated with e.g. sampling, however, it is not necessarily trivial, and usually a simpler expression could be obtained.

For instance, in the case of an atomic intervention on $x_t$ (or any intervention with an empty set of new parents), adjusting using the parents of $x_t$ in the observed BN graph $\mathcal{G}$ is sufficient. Consider estimating $q(x_y)$ under such an intervention when $X_y \notin \mathbf{X}_{\mathbf{PA}_t^{\mathcal{G}}}$. Then, we can obtain a potentially much simpler expression:

$$q(x_y) = q(x_y|x_t) = \int p(x_y|x_t, \mathbf{x}_{\mathbf{PA}_t^{\mathcal{G}}})p(\mathbf{x}_{\mathbf{PA}_t^{\mathcal{G}}})d\mathbf{x}_{\mathbf{PA}_t^{\mathcal{G}}} \tag{37}$$

The last expression in fact matches the ones we obtained for the aspirin intervention earlier: $\int p_{\boldsymbol{\theta}}(y^*|t^*, z^*)p_{\boldsymbol{\theta}}(z^*)dz^*$. Pearl and others have proposed other "adjustment sets" of this sorts, including the *backdoor criterion* and *towards necessity* Peters et al. (2017, Proposition 6.41).

Beyond just obtaining expressions for probability distributions of interest, another consideration comes into play when some of the variables in $\mathbf{X}$ are unobserved. This would mean that, for conditional probability functions in the observed setting $p(\mathbf{x}_A|\mathbf{x}_B)$, the ones for which a subset of variables in either $\mathbf{x}_A$ or $\mathbf{x}_B$ correspond to unobserved variables would not be easily estimable. Hence, one may ask: can we express a conditional probability function in the interventional setting using only the probability density/mass functions of the observed variables (in the observational setting) only? This would clearly be desirable, as any such probability function can be fitted via standard supervised learning; but when is that possible? And how could we obtain such an expression (potentially automatically from the graph)?

The *do*-calculus has been developed to address these questions. We've put its rules in the appendix A.3, or you can find them in Pearl (2009, §3.4) or Peters et al. (2017, §6.7). A particular probability distribution in the interventional world is called *identifiable* if it can be expressed in terms of distribution functions of *observed* variables in the observed world only Peters et al. (2017, p. 119) Pearl (2009, Corollary 3.4.2). The *do*-calculus allows for finding expressions for all identifiable intervention distributions with a repeated application of its rules[13]. Using the *do*-calculus, Tian and Pearl have developed an algorithm that is guaranteed to find all identifiable intervention distributions (Tian & Pearl, 2012), and Schipster and Pearl have developed a graphical criterion on $\mathcal{G}$ for determining identifiability of an interventional distribution (Shpitser & Pearl, 2006).

The choice to include 'calculus' in the name of the *do*-calculus adds a ring of profoundness to it, and inspires curiosity. This has possibly instilled a somewhat warped expectation of what it actually is in some. The *do*-calculus simply builds on the rules of probability theory, the definition of a BN and the definition of interventions in BNs. Its rules are derived from these, and hence any result obtained using the *do*-calculus can be obtained from these underlying rules as well.

---

[13]Which you'd hope seeing as you can do this by using Defs. 2.1 and A.3 and the rules of probability theory.

### J.1. Identifiability

Above, we briefly introduced the concept of identifiability: an interventional conditional probability function (the "causal effect") is *identifiable* whenever it can be expressed in terms of probability functions of the observed variables only Pearl (2009, p. 77). This criterion conforms to a frequentist notion of identifiability wherein we would require that a quantity of interest can be uniquely determined in the infinite observed data limit[14].

Note that, from a Bayesian perspective, identifiability is not a requirement for drawing inferences about a quantity of interest. From a Bayesian perspective, as long as we specify the model in full, there is nothing stopping us conceptually from computing a posterior over any quantity of interest in our model. We can still concern ourselves with whether observations reduce the (epistemic) uncertainty in the quantity we wish to infer. Generally, we won't be able to to reduce the epistemic uncertainty completely in the presence of non-identifiability, even in the infinite data limit, unless we encode further restrictive assumptions through the use of the prior (in effect constraining the model class until there is identifiability). However, even in the presence of non-identifiability, *Bayesian learning* can still occur (defined as whenever the posterior distributions can differ from the prior distributions, reflecting that we have updated our beliefs based on the data (Palomo et al., 2007)). We recommend (MacKay, 2003, § 28) for a useful discussion on this topic.

## K. What is a Causal Bayesian Network?

Here, we expand on the discussion of the different definitions of a *Causal Bayesian Network* found in the literature.

In the main body of the paper, we recommended viewing the term *Causal Bayesian Network* as a — potentially useful — piece of terminology or jargon. It has not additional mathematical meaning beyond that conveyed by the term Bayesian Network. The term indicates that, when defining a *Causal* Bayesian Network over the observed variables, we *intend* to use the operation in Def. 2.1 to define an interventional distribution from it, and that this obtained distribution will reflect our assumptions about the intervened-upon setting. Whereas in Example 1 we verbally described how we interpret the variables $\mathbf{Z}^*, \mathbf{T}^*, \mathbf{Y}^*$ in the intervened-upon world, the interpretation of the variables resulting from applying the intervention operation Def. 2.1 can be implicitly implied by terming the original Bayesian Network *'causal'*.

This is not too far off from how Pearl (2009, p. 23) would define Causal Bayesian Networks. Pearl's definition (effectively) starts with a collection of *all* (atomic) 'interventional' distributions and a Bayesian Network, and says that the Bayesian Network is a *Causal Bayesian Network* for the collection of interventional distributions if applying the rule in Def. 2.1 to the unintervened distribution matches the respective interventional distribution.[15] With that definition, a Bayesian Network is only a *causal* Bayesian Network with respect to something — the collection of interventional distributions. To establish that a Bayesian Network is *causal*, we need to specify what the set of interventional distributions is. If we leave it implicit, we effectively just say that the Bayesian Network and the rule in Def. 2.1 is a valid shorthand for those interventional distributions.

On the other hand, Peters et al. (2017, Def. 6.32) define a Causal Graphical Model (their name for a Causal Bayesian Network) as effectively a Bayesian Network to which the rule in Def. 2.1 can be applied. As we mentioned in the main text, this definition is, mathematically, not adding anything to the definition of a Bayesian Network. The underlying mathematical object (graph and random variables) is the same, and we could just as well apply the intervention operation to a Bayesian Network as well as the *Causal Graphical Model*.

Some would define Causal Bayesian Networks as Bayesian Networks in which the edges are 'causal'. However, one quickly finds out that this has to be a claim about the interpretation of the mathematical object at hand, rather than a definition of it.

Among all the definitions, the one given by Cohen (2022) is perhaps the clearest: *"A Causal Bayesian Network is a Bayesian Network with the word 'causal' prepended"*. Although intended to be humorous, as a mathematical definition, it just might be the most rigorous as well.

---

[14]As, under some reasonable assumptions, we would be able to infer all the distribution functions for all observed variables in the infinite data limit.

[15]Pearl's definition actually has one slight difference to how we described it. One would start with the collection of all interventional distributions (including the unintervened one) and a graph. Then one can say that the *graph* is a Causal Bayesian Network consistent with the interventional distributions if applying the rule in Def. 2.1 to the Bayesian Network obtained by combinig the unintervened distribution and the graph matches the respective interventional distribution. The difference to how we described it is inconsequential, as the the unintervened distribution and the graph can be together comprised into a Bayesian Network at the outset.

## L. Viewing twin model inference as two-stage inference in two separate models

There is an alternative way to arrive at the inference procedure we followed in Example 2. Namely, just as we could equivalently frame our twin model procedure in the interventional setting as inference in two separate models (the observed and the interventional), an analogue perspective holds for counterfactuals. Recall from the graphical model that the counterfactual variables $T^*$, $Y^*$ are independent of the variables in the observed setting conditioned on $Z$ (and model parameters $\boldsymbol{\theta}$). We could define a model over a single set of counterfactual variables — $Z^*$, $T^*$, $Y^*$ — and set the prior on $Z^*$ in that model to equal the posterior on $Z$ in the observed world:

$$p_{\boldsymbol{\theta}}(Z^* = z) = p_{\boldsymbol{\theta}}(z|t, y)$$

If we then run inference in that model, say conditioned on $T^* = t^*$, we would obtain:

$$\mathbb{E}_{\boldsymbol{\theta} \sim p(\boldsymbol{\theta}|\mathcal{D}, t, y)}\left[p_{\boldsymbol{\theta}}(y^*|t^*)\right] = \mathbb{E}_{\boldsymbol{\theta} \sim p(\boldsymbol{\theta}|\mathcal{D}, t, y)}\left[\int p_{\boldsymbol{\theta}}(y^*|t^*, z^*) \underbrace{p_{\boldsymbol{\theta}}(Z^* = z)}_{= p_{\boldsymbol{\theta}}(z|t,y)} \, dz\right] \tag{38}$$

This is the same expression as the one we obtained using the joint model in Equation (6). Hence, we can completely equivalently view the joint model approach we have outlined for the aspirin example as the posterior in the observed world becoming the prior for a counterfactual world model.

## M. Other reasons to not share all the noise variables in counterfactual inference

Deciding whether to share all the noise variables between the observed and the counterfactual worlds gets even more ambiguous once we consider noise variables that have been added to account for modelling error, rather than to represent actual stochasticity in the data. This is a common practice in machine learning; even if the problem is known or assumed to be deterministic, we would often specify a small observational noise to: **1)** deal with models of finite capacity, or **2)** make the likelihood continuous with respect to model parameters to allow for gradient-based optimisation. If that is the case, sharing the noise variable between the observed and counterfactual worlds is hard to link to an intuitive interpretation.

## N. Counterfactual inference in the aspirin example sharing all the noise variables

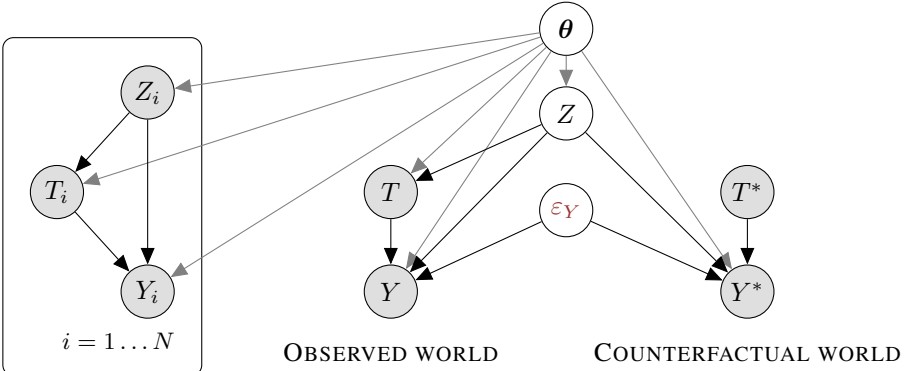

*Figure 9.* Graphical model for counterfactual inference in the aspirin example with the noise variable $\varepsilon_Y$ shared between the two settings.

We can consider what happens when we share $\varepsilon_Y$ across the observed and counterfactual settings in Example 2. For the particular parameterisation on $\varepsilon_Y$ that we have chosen, $Y = \frac{Z^b}{T^c}\varepsilon_Y$, $Y$ is a deterministic function of $Z$, $T$ and $\varepsilon_Y$ (conditioned on $\boldsymbol{\theta}$). Hence, by sharing both $Z$ and $\varepsilon_Y$ we could uniquely identify the counterfactual value of $Y^*$ for any given $T^*$. To see this, we can write (assuming $Y^*$ has the same functional dependence on $\varepsilon_Y, Z, T^*$):

$$Y^* = \frac{Z^b}{(T^*)^c}\varepsilon_Y = \left(\frac{Z^b}{T^c}\varepsilon_Y\right)\frac{T^c}{(T^*)^c} = Y\frac{T^c}{(T^*)^c} \tag{39}$$

which, assuming point-estimate of model parameters $\boldsymbol{\theta}$, uniquely identifies the value of $Y^*$ for any given set of $Y, T, T^*$.

### N.1. Equivalence to the Structural Causal Model Approach

It is illustrative to verify that the same answer follows if we follow the standard Structural Causal Model approach.

Since the joint distribution of $\mathcal{E}, T, Y$ (where $\mathcal{E}$ are the latent "noise" variables) does not have a density, and since strictly following the "abduction, action, prediction" definition of counterfactuals in (Peters et al., 2017) requires computing a conditional probability distribution over $\mathcal{E}$ conditioned on $(T, Y)$, the measure-theoretic details become necessary. This is another advantage of viewing counterfactuals as a single twin model: we were able to avoid these awkward inference details by not having to manifest a conditional probability.

Let $\mathcal{E} = (\mathcal{E}_Z, \mathcal{E}_T, \mathcal{E}_Y)$ be independent random variables (we implicitly assume a probability space $(\Omega, \mathcal{F}, P)$ throughout) with distribution implied by the density:

$$p_{\mathcal{E}}(\epsilon) = \log \mathcal{N}(\epsilon_Z; \mu_Z, \sigma_Z^2) \log \mathcal{N}(\epsilon_T; 0, \sigma_T^2) \log \mathcal{N}(\epsilon_Y; 0, \sigma_Y^2)$$

Let random variables $(Z, T, Y)$ be defined with structural assignments $(f_Z, f_T, f_Y)$:

$$Z = f_Z(\mathcal{E}_Z) = \mathcal{E}_Z \qquad T = f_T(Z, \mathcal{E}_T) = Z^a \mathcal{E}_T \qquad Y = f_Y(T, Z, \mathcal{E}_Y) = \frac{Z^b}{T^c} \mathcal{E}_Y.$$

Then, $\langle S, \mathcal{G}, p_{\mathcal{E}} \rangle$ is an SCM on the random variables $(Z, T, Y)$ with:

- $\mathcal{G}$ a directed graph on $(Z, T, Y)$ with edges $\{(Z, T), (Z, Y), (T, Y)\}$

- $p_{\mathcal{E}}$ as defined above

- $S = (f_Z, f_T, f_Y)$

Conditioned on any observation of $(T, Y)$, we would construct a counterfactual SCM with exogenous variables $\mathcal{E}^* = (\mathcal{E}_Z^*, \mathcal{E}_T^*, \mathcal{E}_Y^*)$ distributed according to a version of conditional probability distribution (Billingsley, 1995, p. 439):

$$P[\mathcal{E}^* \in \cdot] := P[\mathcal{E} \in \cdot || T, Y]$$

and random variables $(Z^*, T^*, Y^*)$ defined with the following intervened-upon structural equations with an atomic intervention on $T$:

$$Z^* = f_Z(\mathcal{E}_Z^*) = \mathcal{E}_Z^* \qquad T^* = t^* \quad (t^* \text{ is a constant}) \qquad Y^* = f_Y(T^*, Z^*, \mathcal{E}_Y^*) = \frac{(Z^*)^b}{(T^*)^c} \mathcal{E}_Y^*.$$

Note, that $P[\mathcal{E}^* \in \cdot]$ is implicitly a conditional distribution (a function of $\omega \in \Omega$ that's measurable $\sigma((T, Y))$, but the dependence is notationally suppressed).

Then, for every value of $\omega \in \Omega$, $\langle S^*, \mathcal{G}^*, P[\mathcal{E}^* \in \cdot] \rangle$ is a counterfactual SCM on $(Z^*, T^*, Y^*)$, where $S^* = (f_Z, t^*, f_Y)$ and $\mathcal{G}^*$ is directed graph on $(Z^*, T^*, Y^*)$ with edges $\{(Z^*, Y^*), (T^*, Y^*)\}$. In this SCM:

$$P[Y^* = y^*] = P[\frac{(Z^*)^b}{(t^*)^c}\mathcal{E}_Y^* = y^*] = P[\frac{(\mathcal{E}_Z^*)^b}{(t^*)^c}\mathcal{E}_Y^* = y^*]$$

△Get into the form of a probability distribution for $\mathcal{E}^*$

$$= P[\boldsymbol{\mathcal{E}^*} \in \{\boldsymbol{\epsilon} : \frac{(\epsilon_Z)^b}{(t^*)^c}\epsilon_Y^* = y^*\}]$$

△Substitute in the definition of probability distribution for $\mathcal{E}^*$

$$= P[\boldsymbol{\mathcal{E}} \in \{\boldsymbol{\epsilon} : \frac{(\epsilon_Z)^b}{(t^*)^c}\epsilon_Y^* = y^*\}||T,Y]$$

$$= P\left[\left[\frac{(\mathcal{E}_Z)^b}{(t^*)^c}\mathcal{E}_Y = y^*\right]||T,Y\right]$$

$$= P\left[\left[\frac{(T)^c}{(t^*)^c}\frac{(\mathcal{E}_Z)^b}{(T)^c}\mathcal{E}_Y = y^*\right]||T,Y\right]$$

$$= P\left[\left[\frac{(T)^c}{(t^*)^c}\frac{(Z)^b}{(T)^c}\mathcal{E}_Y = y^*\right]||T,Y\right]$$

$$= P\left[\left[\frac{(T)^c}{(t^*)^c}Y = y^*\right]||T,Y\right]$$

△Since $\frac{(T)^c}{(t^*)^c}Y = \frac{(T)^c}{(t^*)^c}Y$ is true on the entire sample space $\Omega$:

$$= \begin{cases} P[\Omega||T,Y] & \text{if } y^* = \frac{(T)^c}{(t^*)^c}Y \\ P[\emptyset||T,Y] & \text{otherwise} \end{cases}$$

$$= \begin{cases} 1 & \text{if } y^* = \frac{(T)^c}{(t^*)^c}Y \\ 0 & \text{otherwise} \end{cases},$$

where the last equality holds with probability 1 by the properties of any conditional probability (Billingsley, 1995, eq. (33.27) and (33.28)).

## O. Non-identifiability – different latent representations can yield different counterfactuals

There is another complication with trying to share an unobserved variable between the counterfactual and the observed settings. Let's say that we hypothesise the existence of a latent variable that represents some person or situation-specific properties relating to how the headache duration will be affected by a given dose of aspirin. There are potentially infinitely many parameterisations on such a latent that will give the same observed distribution, but different counterfactual outcomes.

For a concrete example, consider what would happen if the headache duration's ($Y$) dependence on $\varepsilon_Y$ had instead been defined as:

$$Y = \frac{Z^b}{T^c}(\varepsilon_Y)^{\text{sign}(\log T)} \tag{40}$$

The marginal distribution on $(Z, T, Y)$ would be left completely unchanged by this revision; we've left the distributions on $Z$ and $T$ unchanged, and the conditional distribution on $Y$ given $Z, T$ is the same, because both $(\varepsilon_Y)^1$ and $(\varepsilon_Y)^{-1}$ are distributed as $\log \mathcal{N}(0,1)$[16].

Now, consider what would then happen if we shared $\varepsilon_Y$ with the counterfactual world (let's for convenience assume that $Z$ is observed this time). From eq. 40, we could infer the value of the latent: $\varepsilon_Y = \left(\frac{YT^c}{Z^b}\right)^{\text{sign}(\log T)}$. The equation for the

---

[16]To see this, we can write $(\varepsilon_Y)^{-1} = \exp(-1 \log \varepsilon_Y)$ and recall that $\log \varepsilon_Y \sim \mathcal{N}(0,1)$. A standard normal distributed variable multiplied by $-1$ is, however, also standard normal distributed.

counterfactual headache duration would then take the form:

$$Y^* = \frac{Z^b}{(T^*)^c}\left(\varepsilon_Y\right)^{\text{sign}(\log T^*)} = \begin{cases} \frac{Z^b}{(T^*)^c}\varepsilon_Y^1 \\ \frac{Z^b}{(T^*)^c}\varepsilon_Y^{-1} \end{cases} = \begin{cases} \frac{Z^b}{(T^*)^c}\left(\frac{YT^c}{Z^b}\right)^{\text{sign}(\log T)} & \text{if } \log T^* \geq 0 \\ \frac{Z^b}{(T^*)^c}\left(\frac{YT^c}{Z^b}\right)^{-\text{sign}(\log T)} & \text{if } \log T^* < 0 \end{cases}$$

This is compared to the case for the standard formulation ($Y = \frac{Z^b}{T^c}\varepsilon_Y$), where the counterfactual outcome takes the form $Y\frac{T^c}{(T^*)^c}$ as shown in eq. 39. Hence, clearly, the inferred counterfactual quantity would be different for these two formulations.

Without additional assumptions, there are infinitely many ways in which the observed variables can depend on the latent variables that could give different counterfactual outcomes. Hence, returning to the recurring theme, we need to specify these assumptions to be able to do counterfactual inference.

## P. A short causal problem not conveniently represented by a graphical model

Concretely, consider a system in which the number of random variables for one observation depends on another random variable. Take as an example a casino game in which the player throws a dice, and depending on its outcome $D$, they draw $D$ cards $\{C_i\}_{i=1}^D$ from the deck. Then, the sum of the values of the cards drawn is compared to that of a dealer to determine the payout. For a simple six-sided die, we 'could' specify six distinct card nodes $\hat{C}_i$ in a Bayesian Network graph, and augment their sample space to include a special symbol when they aren't drawn. But what if $D$ is an infinite-sided die (a discrete random variable taking on values in $\{0, 1, 2, \ldots\}$ with some probability), and we have an infinite-sized deck? Specifying a graph on an infinite number of nodes seems cumbersome — the Bayesian Network representation for this system seems somewhat contrived. Nonetheless, it's still easy to imagine interventional and counterfactual questions one might ask about this system (e.g. "would I have won had the die rolled a 6?").

## Q. Causal-statistical dichotomy

A quick glance at the causality debate reveals one of the major recurring themes: the proposition that causality is in some sense strictly distinct from statistical analysis. Judea Pearl has repeatedly insisted that these two differ in their goals and approaches, even coming as far as to say *"If I am remembered for no other contribution except for insisting on the causal-statistical distinction, I would consider my scientific work worthwhile"* Pearl (2009, p. 332). In this section, we summarise and comment on the arguments for such a distinction, hopefully addressing and resolving the confusion that has arisen from this claim.

The rationale for the causal-statistical dichotomy claim is grounded in the argument that standard statistical analysis' only aim is to assess properties of a *static* distribution. Judea Pearl proclaims that causal analysis *"goes one step further; its aim is to infer not only the likelihood of events under static conditions, but also the dynamics of events under changing conditions, for example, changes induced by treatments or external interventions, or by new policies or new experimental designs"* Pearl (2009, p. 332). Pearl maintains that there is no more to statistics than inferring or answering questions about the properties of the observational distribution $p(\mathbf{x})$. Once any property of $p(\mathbf{x})$ of interest can be tractably computed with no uncertainty, there is nothing else left to do within the realm of statistics.

This argument, of course, rests on imposing the above constraining definition onto statistics; it relies on accepting that statistics does in fact only deal with observational distributions and static conditions. Many researchers who consider themselves statisticians would, however, disagree with the claim that statistics reduces to inference within the observational distribution, as there is plenty of work within the statistical realm that extends to non-static domains as well. For instance, research in covariate and dataset shift, domain adaptation, off-policy reinforcement learning, or robustness is all about dealing with changes in the distribution from which the variables are drawn.

If one were to consider causal-statistical distinction as a novel terminological proposition rather than a claim about the historical limits of the fields, the nature of the argument becomes clearer. As there is nothing in a distribution function that tells us how it would change if the external conditions were to change, there is a need for additional assumptions to answer these kinds of questions. Andrew Gelman, who has notably criticised Pearl's claims in his review of "The Book of Why" (Gelman, 2019), admits that he "agree[s] [with Judea Pearl] that data analysis alone cannot solve any causal problems. Substantive assumptions are necessary too". One might then argue that these extra assumptions should be called *causal assumptions*, and analysis that goes beyond reasoning about variables distributed according to the observed distribution should be referred to as *causal analysis*. However, the necessity for assumptions is not unique to the causal setting. In

his 2003 book, MacKay writes: "You can't do inference – or data compression – without making assumptions". These assumptions are going to be subjective, but once the human input enters through the design of the hypothesis space, in the probabilistic modelling paradigm, the inference is mechanical. Hence, the fact that causal inference requires subjective assumptions is not a convincing argument for delineating causal inference from statistics. Lastly, as the term *statistical analysis* has historically not been used exclusively to refer to the restricted scope of static distributions, the insistence on the existence of a dichotomy between statistics and causal analysis is somewhat confusing.

In the paper "Causal inference in statistics: an overview" Judea Pearl tempers the terminological separation[17] described above by suggesting a distinction between *associational* and causal concepts instead Pearl et al. (2009). There, he draws a demarcation line by saying that an associational concept is any relationship that can be defined in terms of a joint distribution of observed variables, whereas a causal concept is any relationship that cannot be defined from the observed distribution alone. Although this definition does not rely on unfaithfully limiting the scope of statistics, it does impose the causal trademark onto a broad class of concepts.

---

[17]Perhaps to appease the statisticians to whom the paper is addressed.

