# OpenReview forum: "Position: Probabilistic Modelling is Sufficient for Causal Inference"
_ICML.cc/2025/Position_Paper_Track — ICML 2025 Position Paper Track oral_

### Official Review · Reviewer_xMS4 · 2025-02-16

**Significance:** 3
**Argument Clarity:** 3
**Rating:** 3
**Confidence:** 3

**Questions:**

•	In Equation 1, what is the rationale for using a log-normal distribution for $Z$, $\varepsilon_{T}$, and $\varepsilon_{Y}$? What underlying assumptions are required to justify these distributional choices?

•	In Eq. 2, why is the term $q\_{\boldsymbol{\theta}}(t^*)$ dependent on the parameter $\boldsymbol{\theta}$?

**Discussion Potential:**

4

**Ethical Review Concerns:**

I do not think the presented work has any significant ethical concerns.

**Paper Summary:**

This paper presents the position that probabilistic modeling is useful, underutilized, and underemphasized in causal inference. Specifically, the authors argue that we do not need causal-specific tools or notations (e.g., the $do$-operator introduced by Pearl) to answer causal questions. To support this claim, they analyze the effect of aspirin dosage on headache duration, using causal graphs (or Bayesian Networks) to examine both the intervention effect of dosage variation and counterfactual estimation of headache duration. The results demonstrate that both interventional and counterfactual quantities can be derived purely within a probabilistic modeling framework. Finally, the authors discuss the advantages of this approach, alternative perspectives, and its relationship to Structural Causal Models (SCMs).

**Position:**

Yes

**Position In Title:**

Yes

**Related Work:**

2

**Strengths And Weaknesses:**

**Strengths**:

•	The debate over the necessity of causal-specific tools and notations is a significant issue in causal inference. Resolving this question could substantially lower the entry barrier and reduce the time required for future researchers to engage in causal inference studies.

•	The examples presented in the paper is intuitional and helpful for readers to understand the authors’ main argument.

•	It seems that with correct assumptions, the effect of one variable on another in the causal graph can be quantified and better understood with the probabilistic modeling approach.

**Weaknesses**:

•	In Section 3.1, the authors provide some insights on what variables need to be shared between the observed and the interventional BNs in different contexts. However, these insights appear to be limited to the simple case $T \leftarrow Z \rightarrow Y$ presented in Example 1. It would be valuable for the authors to extend their discussion to more complex scenarios, e.g., when $Z$ is a latent variable and we only get to see an associated proxy variable $X$ (i.e., $ T \leftarrow Z \rightarrow Y $ and $Z \rightarrow X$).

•	In Examples 1 and 2, the simple causal structure $T \leftarrow Z \rightarrow Y$ allows us to get analytical expressions of $\mathbb{E} [Y | \theta]$ and $p(y^{*}|t, y, \mathcal{D})$. However, in real-world applications, the assumed variance distributions of variables may differ, and causal structures can be more complex. Consequently, computing the conditional or marginal distributions of the target variable and estimating the intervention effect can become challenging. It would be beneficial for the authors to further discuss the limitations of the proposed probabilistic framework in such scenarios.

**Support:**

3

---

> ### Author Rebuttal · Authors · 2025-04-01
>
> Thank you for your review!
>
> > [...] authors provide some insights on what variables need to be shared between the observed and the interventional [...] contexts. However, these insights appear to be limited to the simple case  presented in Example 1. It would be valuable for the authors to extend their discussion to more complex scenarios [...].
>
> The high-level point we were trying to make is that choosing which latents to share between the observed and the counterfactual settings can be problem dependent — it's a decision that the practitioner has to make depending on what specific inference question they want answered. The example in section 3.1 was intended as an illustration of what that process might look like. If a more complex example would be useful, we could expand the discussion in Appendix M.
>
> For the example that you've given with an additional proxy variable, do you think much would change in the surrounding discussion of which latent variables to share? I think the discussion about which latents to share might be very similar, since the latents under consideration would more or less be the same, but let me know if I might have missed something that would change in this setting.
>
> > In Examples 1 and 2, the simple causal structure  allows us to get analytical expressions [...]. It would be beneficial for the authors to further discuss the limitations of the proposed probabilistic framework in [more complex real-world scenarios].
>
> Yes, that is a great point. We have written Appendix G precisely to include a discussion on the practical aspects of answering causal inference questions in a probabilistic framework. This appendix is not currently being linked to anywhere from the main text, so **we will make sure to link to it and discuss it appropriately in the main text.**
>
> > In Equation 1, what is the rationale for using a log-normal distribution [...] What underlying assumptions are required to justify these distributional choices?
>
> The log-normal distribution is a variation on the cliché multivariate Gaussian example used in nearly all causality textbooks. We chose it as: 1) the non-negativity of the log-Normal random variables fits the setting (variables represent dose, headache duration, etc.), 2) conditioning is still analytically tractable, making for a good illustrative study, and 3) it's slightly distinct from the examples other authors used in the past.
> There are some reasons to use a log-normal distribution in practice (e.g. multiplicative central limit theorem), but doing so of course comes with many limitations on what kind of effects can be captured – e.g. the not being able to capture multi-modality.
>
> Please let us know if you think any of this rationalé should be included in the paper.
>
> > In Eq. 2, why is the term  dependent on the parameter ?
>
> You're correct in pointing out that it's not, so the subscript is redundant. We'll remove it and on **L155 (right) we will write:** “To represent the belief that we are manipulating the system to assign a specific dose $t^\ast$ independently of $Z^\ast$, we specify $q_\theta(t)= q(t) = \delta(t^\ast − t)$.”

---

> > ### Comment · Reviewer_xMS4 · 2025-04-02
> >
> > Thank you for your responses to my questions. I think the answers most address my concerns. For the additional proxy variable case, I think the latent variables to share can be similar to the case without proxy variable, but the associated Bayesian network might be different, and so does the joint density. Therefore, it might be beneficial to include an additional example.

---

> > > ### Author Response · Authors · 2025-04-06
> > >
> > > Thank you for the feedback. In that case, we'll draft a walk-through of this example up for an appendix. If time permits, we'll try to share it before the end of the discussion period.

---

### Official Review · Reviewer_EqFF · 2025-03-04

**Significance:** 4
**Argument Clarity:** 4
**Rating:** 4
**Confidence:** 4

**Questions:**

Even if the causal questions, e.g., the interventional questions in the first aspirin example, can be answered using existing probabilistic language, doesn't the syntax/vocabulary introduced with the do-operator offer value when discussing causal questions? For example, without the do-operator one would need to use natural language to ask the "expected effect of intervening to assign someone a given dose of aspirin t* on their headache duration on average"; which would be simple to write down using the do-operator.

Continuing, I would like to ask a counterfactual question: if there are no previous results/languages of do-intervention, the structural causal model, and the twin models of SCMs for analysing counterfactuals, would there be the approach of writing down the probabilities of observed BN and the intervened BN as discussed in this paper? Or, wouldn't it be a much more challenging mental effort to write down the twin models?

Additionally, there are some minor grammatical mistakes scattered in the manuscript, I list a few of them in the following:

1. At Page 4, Line 193, there was an incorrect reference to Equation (24). In Equation 24, the derivation refers to Equation 4, which should be Equation 3?
2. Page 6, Line 322, "it might counterintuitive" --> "it might be counterintuitive"
3. Page 20, line 1064, "what if the there is" --> "what if there is"

**Discussion Potential:**

4

**Paper Summary:**

This position paper discusses whether probabilistic language is sufficient to answer causal questions and, therefore, whether Pearl's causal framework is fundamentally different from statistical tools. It discusses how the probabilistic language can be used to answer an intervention question and a counterfactual question, showing that the causal estimands can be derived from the joint probabilities if one writes the observed joint distribution side-by-side with the interventional joint distribution and derives expectations from there. It also discusses the position that in some cases, e.g., counterfactual inference, the assumptions that come with the structural causal model can be overly restrictive in the sense that some information in the already happened event should not be shared to the counterfactual world. Overall, the paper clearly states its position: causal inference can be achieved through probabilistic modelling by defining a model over the observed and interventional/counterfactual distributions, and the probabilistic framework offers flexibility over the causal framework in certain cases.

**Position:**

Yes

**Position In Title:**

Yes

**Related Work:**

4

**Strengths And Weaknesses:**

Strengths:
1. This paper is well-written, with its position clearly stated and well-supported by evidence.
2. The position of this paper relates to a fundamental problem in machine learning, and is likely to inspire further discussion.
3. The related works are appropriately cited and discussed.

**Support:**

4

---

> ### Author Rebuttal · Authors · 2025-04-01
>
> Thanks a lot for a thorough review, and suggestions for polishing the paper!
>
> ### Addressing questions:
> > Even if the causal questions, e.g., the interventional questions in the first aspirin example, can be answered using existing probabilistic language, doesn't the syntax/vocabulary introduced with the do-operator offer value when discussing causal questions?
>
> Yes, absolutely. This is what we tried to convey with: “There are convenient classes of models [...], notational shorthands (e.g. the do-operator), and a machinery developed around them [...].” (L34) Shorthand was intended as a positive term. If different wording would better emphasise that point, please do let us know.
>
> > Continuing, I would like to ask a counterfactual question: if there are no previous results/languages of do-intervention, the structural causal model, and the twin models of SCMs for analysing counterfactuals, would there be the approach of writing down the probabilities of observed BN and the intervened BN as discussed in this paper? Or, wouldn't it be a much more challenging mental effort to write down the twin models?
>
> It seems like a nuanced question. If the question is about a counterfactual history in which the SCM/do-calculus frameworks don't exist: people have been doing very _similar_ things to what's described in the paper before Pearl. The potential outcomes framework, which precedes Pearl's, effectively also starts off by specifying a joint distribution (implicitly, through defining observed and counterfactual random variables). Similarly, off-policy reinforcement learning (RL) is also about computing probabilities in a counterfactual world, and the RL community has been computing (a point we briefly make in Appendix Q). Hence, it seems plausible someone could have figured out how to tackle interventional and counterfactual questions more broadly with Bayesian networks. That being said: having Pearl's framework and insights certainly makes the realisation that one can answer causal inference questions with Bayesian networks much more straightforward.
>
> If the question is about whether one would be able to tackle causal inference questions with a probabilistic framework without first learning Pearl's framework, I think the answer is likely yes: the probabilistc community familiar with Bayesian Networks have been constructing similar models for domain shifts/domain adaptations for decades. Anecdotically, I know many people who find thinking in terms of twin models natural, and vice-versa, many who can only construct them after they already have a Pearl's causal graph in mind.
>
> ---
>
> Also, thank you for pointing out the typos in the paper! We have made the appropriate fixes.

---

### Official Review · Reviewer_Pxv9 · 2025-03-09

**Significance:** 2
**Argument Clarity:** 3
**Rating:** 3
**Confidence:** 2

**Questions:**

No

**Discussion Potential:**

2

**Paper Summary:**

The paper challenges the notion that specialized causal frameworks or notation are necessary for causal inference. Instead, it argues that any causal question can be addressed using standard probabilistic modeling and inference. Through concrete examples, the authors demonstrate how causal questions can be framed probabilistically and reinterpret causal tools as natural extensions of standard inference methods.

**Position:**

Yes

**Position In Title:**

Yes

**Related Work:**

3

**Strengths And Weaknesses:**

I do not wish to comment extensively on the perspective presented in this paper, as probabilistic modeling for causal inference is something statisticians and machine learning researchers engage in daily. The authors argue that complex probabilistic modeling alone suffices for causal inference, including interventional and counterfactual reasoning. However, most researchers acknowledge that the first step in causal inference is formulating a causal estimand as a statistical estimand under certain assumptions. Exploring a unified statistical model that abstracts away causal concepts is technically valid. In my view, it is reasonable to allow different causal frameworks, such as the potential outcomes approach and causal graphs, to coexist, even if they stem from different motivations.

I do not consider myself senior enough to make a high-level judgment on whether causal inference should remain a distinct field or be fully subsumed into probabilistic modeling. I would be interested in hearing other reviewers' perspectives on this matter.

**Support:**

3

---

> ### Author Rebuttal · Authors · 2025-04-01
>
> Thanks for taking the time to review the paper! The summary of the paper is very accurate. With regards to the “strengths & weaknesses”, we think it would be useful for us to clarify what we do and do not try to argue in the paper, particularly in response to the following:
>
> > In my view, it is reasonable to allow different causal frameworks, such as the potential outcomes approach and causal graphs. I do not consider myself senior enough to make a high-level judgment on whether causal inference should remain a distinct field or be fully subsumed into probabilistic modeling.
>
> We are not trying to argue to the contrary. We think there is a lot of shared confusion surrouding the causal graphical framework and its strict necessity when people are first introduced to it; we think the probabilistic approach is useful for people to know about, and it might be the best approach to first learn for some. Even thought probabilistic modelling, in some sense, subsumes causal modelling, we wouldn't want to appear to be making the argument that all research in the field should go through that particular level of abstraction. If there are ways in which we can be more precise with the wording in the paper, do let us know!

---

> > ### Comment · Reviewer_Pxv9 · 2025-04-01
> >
> > Thank you for your response and I have updated my score.

---

> > > ### Author Response · Authors · 2025-04-06
> > >
> > > That's appreciated, thank you for your feedback!

---

### Official Review · Reviewer_hnag · 2025-03-09

**Significance:** 3
**Argument Clarity:** 3
**Rating:** 4
**Confidence:** 4

**Questions:**

Regarding the questions, please refer to my detailed concerns regarding the weaknesses mentioned above.

1. How do you define causality?
2. What is the exact position you advocate for?
3. What is the exact alternative view you are refuting?

**Discussion Potential:**

4

**Paper Summary:**

The authors argue that probabilistic modeling is sufficient to answer causal inference questions. The authors also mention that their position is to express that probabilistic modeling is useful, underutilised and underemphasised. To this end, I am unsure which one should take precedence, as one makes a statement about the relation of probabilistic modeling and causality, and the other is only about the former.

The paper provides very detailed examples and excellent illustrations to demonstrate how probabilistic modeling is sufficient to handle interventions (section 2) and counterfactuals (section 3). Section 4 summarizes the benefits of probabilistic modeling, criticizes the alternative view, and concludes by emphasizing the importance of writing down the probability of everything.

**Position:**

Yes

**Position In Title:**

Yes

**Related Work:**

3

**Strengths And Weaknesses:**

## Strengths
The paper is timely, relevant, and will with a high probability incite a lively debate in the community. I appreciate the authors' effort to consolidate perhaps even opposing fields.
The examples are instructive - though the many integrals and complex graphs of the probabilistic approach (especially in Example 2) can be very confusing given that, as a position paper, the aim is to communicate a more high-level message.
The position (regarding which I will detail my concerns below) is made more nuanced by many detailed observations, including (paraphrased)
- the do-notation provides a concise syntax
- L335: the discussion of the conventional definition of counterfactual
- pinpointing that potentially the biggest difference in causality is syntactic (I take the negation of this, ie, "causality differs from probabilistic modeling semantically" as the steelman version of the alternative view the authors presumably discuss - making it clearer would be helpful though)

## Weaknesses

My biggest concerns is the lack of clarity of the *exact* position, and the potentially imbalanced representation of the authors' view and the alternative, thereby _diminishing the admittedly **very high potential of this work to start a fruitful discussion**_.

### How do you define causality?
This is unclear from your position paper. For example, all the DAGs in your examples can be considered causal, i.e., you are using Causal Bayeisan Networks, thus, actually causal tools. Surely that's not what you intended to say, and I beleive it would improve the manuscript if you could eliminate this point of confusion.

### What is the position?
The bolded text on page 1 makes is unclear what the position is. Given the title (and the paper's reasoning), I assume it is about the sufficiency of statistical tools for causal inference.
In any case, it would be very helpful for the reader to clarify how the "clear, unifying, and completely general" statement comes into the picture. Also, it is unclear what those terms exactly mean in this context. The same holds for the "useful, underutilised and underemphasized" listing - describing *for what* problem, scenario, etc. these quantifiers are supposed to hold would be very helpful
Especially that there are examples when the causal approach  yields a simpler 9thus, one might argue, clearer) solution for a problem, as shown by [Leeb et al., 2025](https://arxiv.org/abs/2502.05085) in their Sec 4.3.

On the notion of generality: considering the quotes of the paper from causality works, even those are not claiming generality for causality. It is a special tool for specific purposes. Thus, arguing about the generality of probabilistic modeling might be a moot point and *could potentially hinder a fruitful discussion, as it can be interpreted as a strawman argument against causality* (even though there are researchers showing deeper connections to probabilistic modeling, see below, in "The view of the causality community")

### The view of the causality community
The discussion of Appendix D shows that the sentiment in causality is more nuanced than suggested by the Pawlowski et al., 2020 quote. Checking the list, many of the quotes pinpoint tendencies, shortcomings, and not *impossibility.* This is not to say that *some* works do so.

Perhaps most importantly, the authors do not showcase the recent advancements in the intersection of probabilistic modeling (particularly, nonlinear Independent Component Analysis) and causal inference. These works, including [(Hyvarinen et al., 2023)](http://arxiv.org/abs/2303.16535) and [(Reizinger et al., 2024)](https://openreview.net/forum?id=k03mB41vyM), explicitly highlight how _without many assumptions in probabilistic modeling correspond to ones in causal inference_. Thus, the identifiability results are identical, only their interpretation is different (e.g., distribution shifts vs interventions). Notably, **this might even be used as evidence for the authors' position that causality is just syntactic sugar**. This also implies, though, that the field of causality made very similar conclusions.


### Remarks
- L147: starting by writing down the assumptions is not unique to probabilistic modeling
- Example 1 and L428 (combining interventional and observational data): works from the causality literature like [(Yao et al., 2024)](http://arxiv.org/abs/2409.02772) and [(Reizinger et al., 2024)](https://openreview.net/forum?id=k03mB41vyM) manage that as well
- 2.3.1, L248: the discussion of the Markov equivalence class is actually an argument *for the richness of causal modeling, isn't it?*
- L374, right column: "this procedure is no different": could you please provide evidence for this statement?
- L434, left column: why is it a problem to have a clear blueprint of "abduction, action, prediction"?

**Support:**

3

---

> ### Author Rebuttal · Authors · 2025-04-01
>
> Thank you for a very thorough review! We appreciate that it seems the reviewer considers the perspective and the contributions in the paper noteworthy, with the primary concerns pertaining to the framing and presentation of the argument. It's important to us to represent the position, the alternatives, and the state of the discourse in the field in a balanced and precise way, and we'd be keen to work with the reviewer to improve on that front.
>
> > How do you define causality? [...] What is the position?
>
> The position we argue is primarily about the necessity of using the _causal graphical framework_ — the tools introduced by Pearl specifically for tackling causal problems (SCMs, do-calculus and causal networks) — for causal inference. The main argument is that one can tackle these problems with the tools of probabilistic modelling — such as Bayesian Networks — that predate Pearl's causal graphical modelling.
>
> > i.e., you are using Causal Bayesian Networks, thus, actually causal tools.
>
> The graphical models in Figures 1, 2 & 3 are actually only Bayesian Networks (as defined in Appendix A.2), not Causal Bayesian Networks. We don't rely on any properties of Causal Bayesian Networks until we explicitly introduce, define and use them in Sections 2.3 & 3.2 for comparison.
>
> > it would be very helpful for the reader to clarify how the "clear, unifying, and completely general" statement [and "useful, underutilised and underemphasized"] come into the picture. Also, it is unclear what those terms exactly mean in this context.
>
> It's true that there are few (if any) examples of _active_ opposition to the argument that this perspective is useful or underutilised. What we wanted to say with these statements is that resolving the surrounding debate on the _validity_ of probabilistic modelling for causal inference is _impactful_.
>
> We'll try to break-down what we intended with each term:
> - “clear & useful” - we wanted to convey that being able to solve causal inference problems with probabilistic modelling tools is not a technicallity; i.e. that you can do this in some obscure, difficult-to-understand way.
> - “unifying” - this is a reference to the point made later-on that many causal tools can be viewed “syntactic sugar” in the probabilistic framework.
> - “completely general” - this is w.r.t. solving interventional and coutnerfactual inference problems that can be tackled with the causal graphical modelling framework. It's a good point that we could be more specific: we'll change “completely general” -> ”general for answering causal inference questions”.
> - “useful, underutilised and underemphasised” - with this, we mainly wanted to say that people in the field would benefit from being aware of the connections between probabilistc modelling and causality (i.e. the view of the causal framework that we are presenting). I believe “useful”, we argue in the same paragraph. “underemphasised” is to say that few people — both in causal and probabilistic communities — seem to be aware that this a valid way of tackling causal inference problems. **We could perhaps be clearer by instead saying: “the probabilistic modelling approach is useful and underrecognised”**.
>
> > on the notion of generality: considering the quotes of the paper from causality works, even those are not claiming generality for causality.
>
> The quotes are meant to highglight _seeming_ disagreement with the main premise of the paper: that you _can_ anser interventional and counterfactual inference questions with probabilistic conditioning alone. All the quotes could be read as voicing a sentiment to the contrary (depending on how you interpret “statistical”, as we discuss in Appendix Q). They are not intended to showcase the discourse around generality of the probabilistic or causal frameworks for solving causal questions, as there is little written on that topic.
>
> You're right in that this is currently not clear, and that the quotes need to be contextualised better. We'll make sure that for each quote we surround it with context on what we believe the alternative viewpoint that the quote appears to support is, and to bold the most relevant parts of the quote as well.  We think that would also help with making explicit the fact that some quotes do voice a more nuanced opinion.
>
> > the authors do not showcase the recent advancements in the intersection of probabilistic modeling (particularly, nonlinear Independent Component Analysis) and causal inference
>
> I think that's a good point, and we'd agree that it seems to strengthen the value of the argument that causality in syntactic sugar, and it's useful for the field to be aware of that as showcased by the great work done at the intersection of the two fields. Section 4.1 already has some examples of works that could be considered at the interesection of causality and probabilistic modelling, and we will try to articulate that overall point while citing (Hyvarinen et al., 2023) and (Reizinger et al., 2024) there as well.

---

> > ### Comment · Reviewer_hnag · 2025-04-02
> >
> > Thank you for your detailed response to my questions and concerns!
> >
> > > The quotes are meant to highglight seeming disagreement with the main premise of the paper: that you can anser interventional and counterfactual inference questions with probabilistic conditioning alone. All the quotes could be read as voicing a sentiment to the contrary
> >
> > I presume that you would like to convince the causal community as well that their specific tools (though sometimes optimal) are not necessary. Thus, it would be useful to avoid any interpretation of your position that is an outright dismissal of the causal toolbox. Especially since this is not prevalent in the field. **You should interpret these statements in the strongest possible form to show your goodwill.**
> > For, in my review, I interpret your position in such a way (steelman: you say that the causal toolbox is not necessary, but it is not useless), though one might argue that there is a quite different interpretation (strawman: causality is useless). Please do the same; otherwise, your work might not live up top its very high discussion potential.
> >
> > > Section 4.1 already has some examples of works that could be considered at the interesection of causality and probabilistic modelling, and we will try to articulate that overall point while citing (Hyvarinen et al., 2023) and (Reizinger et al., 2024) there as well.
> >
> > I disagree. You have only a few citations in Sec 4.1, and only one (von Kügelgen et al., 2023) which can be considered part of the modern causal machine learning community. The ICA perspective is totally missing, and so are the recent integrative works at the intersection of these two fields. My point was not about those exact citations, it was more about a meta point: **if you want to convince a community, a very convincing way is to use their own papers to do so** - this is not the only way though. Then, you cannot be ``accused of" criticising something as an outsider, which generally hinders discussion. To be clear, what I want is for you to succeed in initiating this discussion in the most diplomatic and effective way possible. For this, I deem that more effort is required.
> >
> > I am looking forward to your response. I will reconsider my score based on that.
> >
> >
> >
> > **EDIT:**
> >
> >
> > Thank you so much for the clarifications! You got my intent right, i.e., contrasting the strongest versions of both views would be, in my opinion, the most fruitful way to start a constructive discussion. A few examples would be very nice, though I can already tell from your response that you made the effort to provide a high-quality response to my concerns (I like the paragraph you plan to include in 4.1). With this, all my concerns are cleared, and am looking forward to reading the updated version.
> >
> > I increased my score to 4.

---

> > > ### Author Response · Authors · 2025-04-06
> > >
> > > > I interpret your position in such a way (steelman: you say that the causal toolbox is not necessary, but it is not useless), though one might argue that there is a quite different interpretation (strawman: causality is useless).
> > >
> > > Yes, the steelman is certainly what we intended. We can add this sentence explicitly in the introduction: “We claim the causal toolbox is not necessary, but not that it's useless.”
> > >
> > > > Please do the same [for opposing viewpoints]
> > >
> > > Am I correctly interpreting that what you mean is that we should be equaly charitable with interpreting “opposing” viewpoints and quotes? If so, that's a good point. Later in the paper we do make an argument relating to the quotes in Appendix D that, even if they do voice a sentiment that can be interpreted in a completely correct way, they can be easily miconstrued, especially by newcommers to the field (as we argue in Appendix Q).  We agree that it would be productive if, in Appendix D, we discussed for each of quote both a steelman interpretation — in many cases the one the authors likely intended — as well as a proverbial strawman. **We will frame the discussion in Appendix D in this way**. If it would be helpful to see an excerpt of what this discussion will look like for some select quotes, please ask, and we will respond with examples.
> > >
> > > > The ICA perspective is totally missing
> > >
> > > The main overlap between the ICA works and causality, to us, seems to be the concern with identifiability, which we do highlight in the paper (Appenix J.1, linked at L411). It's a fair point that would likely be helpful to reference and point to some of Hyvärinen's and Khemakhem's works on identifiability of nonlinear latent variable models, the methods from which have been ported to look at identifiability of causal representations under various assumptions. See below for more details.
> > >
> > >
> > > > You have only a few citations in Sec 4.1, and only one (von Kügelgen et al., 2023) which can be considered part of the modern causal machine learning community. [...] if you want to convince a community, a very convincing way is to use their own papers to do so
> > >
> > > How about something like the following, in addition to what's already in Sec. 4.1 (we'll tighten it up, this is a first draft):
> > >
> > > In our view, there is plenty of potential for useful work at the intersection of causality and probabilistic modelling. In recent history, that has certainlty been the case. For instance, [(Mossé et al., 2023)](https://arxiv.org/pdf/2111.13936) formally show that causal queries are always reducible to purely probabilistic queries to prove that causal and probabilistic languages have equal computational complexity. The movement to study identifiability in deep learning broadly (see Appendix J.1) has been heavily propelled and influenced by the causal and Independent Component Analysis (ICA) communities (see, for example, the plethora of references to causality and ICA in [(Locatello et al., 2019)](https://arxiv.org/pdf/1811.12359)). For instance, [(Kügelgen et al., 2022)](https://arxiv.org/pdf/2106.04619) study identifiability in a strictly probabilistic model (their Figure 1), but are evidently inspired by identifiability concerns in causality and draw connections between their probabilistic model and interventions in a causal one. This is also the case for [(Locatello et al., 2020)](https://proceedings.mlr.press/v119/locatello20a/locatello20a.pdf), who even explicitly write down the joint distirbution over all the variables of interest as recommended in this paper. Vice-versa, the works on nonlinear ICA [(Hyvärinen,2018;](https://arxiv.org/pdf/1805.08651)[Hyvärinen et al., 2023)](https://arxiv.org/pdf/2303.16535) have led to plenty of novel results on identifiability of causal representation learning [(Hyvärinen et al., 2023)](https://arxiv.org/pdf/2302.02672), [(Kügelgen et al. 2023)](https://proceedings.neurips.cc/paper_files/paper/2023/file/97fe251c25b6f99a2a23b330a75b11d4-Paper-Conference.pdf). [...] Understanding that causal tools can be seen as syntactic shorthands for specifying joint distributions is, in our view, productive towards these endavours.

---

### Decision · Program_Chairs · 2025-04-30

**Decision:**

Accept (oral)

**Comment:**

There is a consensus among the expert reviewers that the paper's positions that **causality is just syntactic sugar** and that **probabilistic modeling is sufficient for causal inference** are likely to generate fruitful discussion within the causal inference and Causal ML community. The authors have engaged with the reviewers and are willing to improve the paper based on the reviewers' feedback.